# How Does the Latitude of Stratospheric Aerosol Injection Affect the Climate in UKESM1?

Matthew Henry[1], Ewa M. Bednarz[2,3,4], and Jim Haywood[1]

[1]Department of Mathematics, University of Exeter, Exeter, UK
[2]Cooperative Institute for Research in Environmental Sciences (CIRES), University of Colorado Boulder, Boulder, CO, USA
[3]NOAA Chemical Sciences Laboratory (NOAA CSL), Boulder, CO, USA
[4]Sibley School of Mechanical and Aerospace Engineering, Cornell University, Ithaca, NY, USA

**Correspondence:** Matthew Henry (m.henry@exeter.ac.uk)

**Abstract.** Stratospheric Aerosol Injection (SAI) refers to a climate intervention method by which aerosols are intentionally added to the lower stratosphere to enhance sunlight reflection and offset some of the adverse effects of global warming. The climate outcomes of SAI depend on the location, amount, and timing of injection, as well as the material used. Here, we isolate the role of the latitude of $SO_2$ injection by comparing different scenarios which have the same global-mean temperature target, altitude of injection, and hemispherically symmetric injection rates. These are: injection at the equator (EQ), and injection at 15°N and S (15N+15S), at 30°N and S (30N+30S), and at 60°N and S (60N+60S). We show that injection at the equator leads to a substantial undercooling of the Arctic, and to a significant reduction in tropical precipitation, reductions in high-latitude ozone, tropical lower stratospheric heating, and strengthening of the stratospheric jets in both hemispheres. Additionally, we find that the most efficient injection locations are the subtropics (15 and 30°N and S), although the 60N+60S strategy only requires around 30% more $SO_2$ injection for the same amount of cooling; the latter also leads to much less stratospheric warming but only marginally increases high-latitude surface cooling. Finally, while all the SAI strategies come with trade-offs, our work shows that the 30N+30S strategy is a good candidate strategy for an inter-model comparison, and is easier to implement than a multi-latitude controller algorithm.

## 1 Introduction

Stratospheric Aerosol Injection (SAI) refers to a climate intervention method by which aerosols (or their gaseous precursors) are added to the lower stratosphere to reflect a small portion of sunlight and thus offset some of the adverse effects of global warming. Previous studies showed that injection at the equator leads to over-cooling of the equator relative to the poles and a reduction in tropical precipitation (Visioni et al., 2021; Jones et al., 2022; Wells et al., 2024). An alternative strategy was developed where injection occurs at different latitudes in the stratosphere (15° and 30°N and S), which enables a control of not only global-mean surface temperature, but also interhemispheric and equator-to-pole temperature gradients (Kravitz et al., 2017; Tilmes et al., 2018a; Richter et al., 2022; Henry et al., 2023). Both Fasullo and Richter (2022) and Henry et al. (2023) showed that the latitudinal distribution of sulphur dioxide ($SO_2$) emission depends both on the model physics and the background scenario. In order to calibrate the controller algorithm, which determines the injection rates at each latitude, the

response to fixed single-point $SO_2$ injection at a range of latitudes was compared in multiple models (Visioni et al., 2023a; Bednarz et al., 2023c).

Previous work demonstrated that the climate outcomes of SAI depend on the strategy used. Using the CESM(WACCM) (Community Earth System Model with the Whole Atmosphere Chemistry Climate Model as its atmospheric component) model, a few studies have systematically varied the altitude, latitude, season, and amount of $SO_2$ injected to isolate the climate effects of these choices. Lee et al. (2023) compared two SAI simulations with a different altitude of injection and the same temperature target. The authors found that a higher-altitude injection substantially increases the lifetime of $SO_2$ and sulfate aerosols and reduces stratospheric moistening, thus increasing the injection efficiency (as measured by the amount of cooling per Tg $SO_2$ injected). The contribution of the aerosol lifetime effects to the injection efficiency was found to be five to six times larger than that of the water vapor feedback. Zhang et al. (2024) varied the latitude of injection using a set of hemispherically symmetric injection strategies and found that both the equatorial injection strategy and the injection at 60°N and S require more $SO_2$ injection to satisfy the same global-mean temperature goal compared to the injection at either 30°N and S or 15°N and S. Furthermore, injecting at 60°N and S led to an extra 1.5K cooling in the Arctic in that model, though it is worth noting that their polar strategy differed from the other three in that the injection happened only in the spring of each hemisphere and at a lower altitude (i.e. 15 km instead of 21.5 km). Bednarz et al. (2023a) used the same dataset as Zhang et al. (2024) and analysed the effect of changing the latitude of injection on the atmospheric circulation and ozone responses, showing substantial differences in these aspects under different SAI strategies. Additionally, Bednarz et al. (2023b) systematically varied the amount of cooling to maintain temperatures at 0.5 to 1.5 degrees above preindustrial temperatures and show that nonlinear changes can occur in the high-latitude circulation and ozone responses. Finally, previous work has shown that changing the season of injection may impact regional climate outcomes (Visioni et al., 2019), and the efficiency in cooling per Tg $SO_2$ is increased when injection is limited to spring when injecting at high Northern latitudes and at 15km (Lee et al., 2021).

Looking into the future, the next set of Geoengineering Model Intercomparison Project (GeoMIP) simulations, "G6-1.5K-SAI", will consist of symmetric injections at 30°N and S and will aim to control the global-mean temperature only (Visioni et al., 2023b). The simpler implementation relative to the four-latitude controller algorithm should enable more climate modelling centres to contribute to the intercomparison, as it will be part of the Coupled Model Intercomparison Project (CMIP) Assessment Report 7 (AR7) Fast Track set of simulations. A more thorough explanation for the choice of scenario and strategy is given in Visioni et al. (2023b).

It is important to analyse the strategy-dependence of SAI in a different Earth System Model to evaluate the robustness of the conclusions drawn from the CESM(WACCM) studies. In this paper, we systematically compare simulations with different latitudes of annually-fixed $SO_2$ injections at 22 km using the United Kingdom Earth System Model 1 (UKESM1), and compare the effects on the surface climate and stratospheric impacts. We first describe the model and simulations performed (Section 2), and then discuss the resulting tropospheric (Section 3.1) and stratospheric (Section 3.2) impacts before summarizing and concluding the study (Section 4).

## 2 Methods

The set of simulations presented in this paper use UKESM1 (Sellar et al., 2019). The physical atmosphere-land-ocean-sea ice model used is HadGEM-GC3.1 (Kuhlbrodt et al., 2018), which uses the Met Office Unified Model (UM) as its atmospheric component. The resolution of the UM is 1.875° longitude by 1.25° latitude resolution, with 85 vertical levels and a model top at 85 km. The chemistry model is the United Kingdom Chemistry and Aerosol (UKCA) chemistry model (Mulcahy et al., 2018; Archibald et al., 2020), which has troposphere-stratosphere chemistry and coupling to a multi-species GLOMAP modal aerosol scheme (Mann et al., 2010). A more detailed description of the UKESM1 model configuration used for this paper is given in Jones et al. (2022).

Table 1 gives an overview of the different sets of simulations with the number of members, simulation objective (i.e. target), injection latitude, and where the set of simulations was first presented (Reference). The baseline set of simulations follows the middle-of-the-road greenhouse gas emission scenario, the Shared Socioeconomic Pathway 2-4.5 (SSP2-4.5), and has five ensemble members. The SSP2-4.5 simulations are one of UKESM1's core simulations carried out as part of the sixth phase of the Coupled Model Intercomparison Project (CMIP6, Sellar et al. (2019)). We compare new SAI simulations (described below) to a set of simulations which was previously presented in Henry et al. (2023) called "Assessing Responses and Impacts of Solar climate intervention on the Earth System", and is denoted ARISE-SAI-1.5 (Richter et al., 2022; Henry et al., 2023). The ARISE-SAI-1.5 simulations have $SO_2$ injection at 21.5 km and four latitudes: 15°N, 15°S, 30°N, and 30°S. The injection at each latitude is updated yearly by an algorithm to maintain the global-mean temperature (T0) as well as the equator-to-pole (T1) and interhemispheric (T2) temperature gradients at the target values; these correspond to the mean over the 20-year period (2014-2033) during which the global-mean surface temperatures value in UKESM1 exceeds its preindustrial value by 1.5K (Henry et al., 2023). The values for T0, T1, and T2 are 288.06K, 0.54K, and -6.05K respectively, and the equations for T1 and T2 are defined in Kravitz et al. (2017) (their equation 1). The four new sets of SAI simulations presented in this paper only aim to maintain the global-mean temperature (T0) at the same target value via $SO_2$ injection at 21.5 km and at either: the equator (EQ), the pair of 15°N and 15°S latitudes (15N+15S), the pair of 30°N and 30°S latitudes (30N+30S), or the pair of 60°N and 60°S latitudes (60N+60S). All SAI simulations use SSP2-4.5 as their background greenhouse gas emission scenario and inject aerosols continuously throughout the year. The implementation of SAI starts in 2035 and lasts for 35 years. Figure 1 shows the global-mean surface temperature for the ensemble-mean of the SSP2-4.5 simulations and the five SAI simulation sets. The EQ strategy does not quite reach its global-mean temperature target, which may be due to the parametrization of the controller algorithm and the relative inefficiency of increasing $SO_2$ emission at the equator.

## 3 Results

### 3.1 Large-scale tropospheric and surface impacts

Figure 2 summarises some key features of the climate response to SAI for the different latitudes of injection. The latitudinal structure of the increase in the stratospheric (550nm) aerosol optical depth (AOD) averaged over 2050-69 is consistent with

**Table 1.** Summary of simulation ensembles.

| Type | # Members | Target = PI+1.5C | Injection Latitude | Reference |
|------|-----------|------------------|-------------------|-----------|
| SSP2-4.5 | 5 | N/A | N/A | Sellar et al. (2019) |
| ARISE-SAI-1.5 | 5 | T0,T1,T2 | 15°N/S and 30°N/S | Henry et al. (2023) |
| EQ | 3 | T0 | Equator | Here |
| 15N+15S | 3 | T0 | 15°N/S | Here |
| 30N+30S | 5 | T0 | 30°N/S | Here |
| 60N+60S | 3 | T0 | 60°N/S | Here |

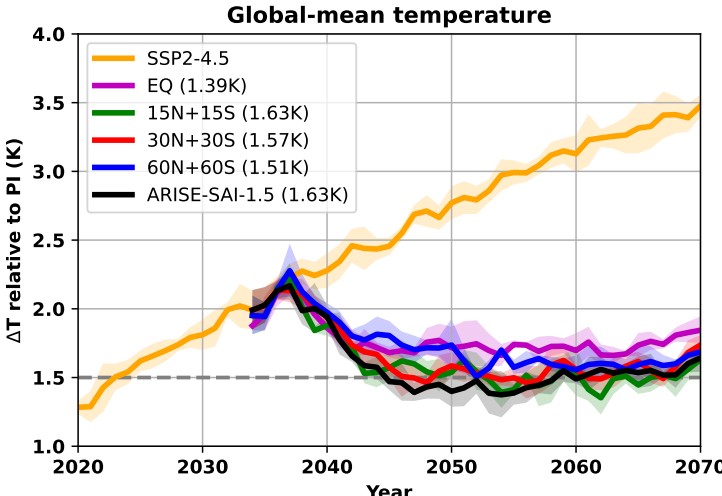

**Figure 1.** Global-mean ensemble-mean surface temperature for SSP2-4.5 (yellow) and each SAI strategy. The dashed grey line represents the global-mean temperature target for all SAI strategies. The global-mean temperature change for 2050-69 relative to SSP2-4.5 is included in the legend for the SAI strategies. The shading represents +/- one standard deviation of each ensemble.

each strategy's injection location (panel a), with the change in AOD maximising near the latitude of injection for the EQ and
15N+15S strategies, and generally poleward of the injection latitude for the 30N+30S and 60N+60S strategies. The injections
in the ARISE-SAI-1.5 simulation are partitioned approximately equally between 30°S, 15°N, and 30°N at the end of the
simulations (Henry et al., 2023). The confinement of aerosols to within the tropical regions by the the so-called "tropical
pipe" is clearly evident in figure 2a and is significantly stronger for UKESM1 compared to other models, as evidenced by the
comparison of single point injections across models in Visioni et al. (2023a) (their figure 2h). The equatorial peak in AOD is
also consistent with simulations presented in Wells et al. (2024) which had a different background scenario and target state.
Note that the baseline stratospheric AOD under SSP2-4.5 is three orders of magnitude smaller than the changes under SAI.

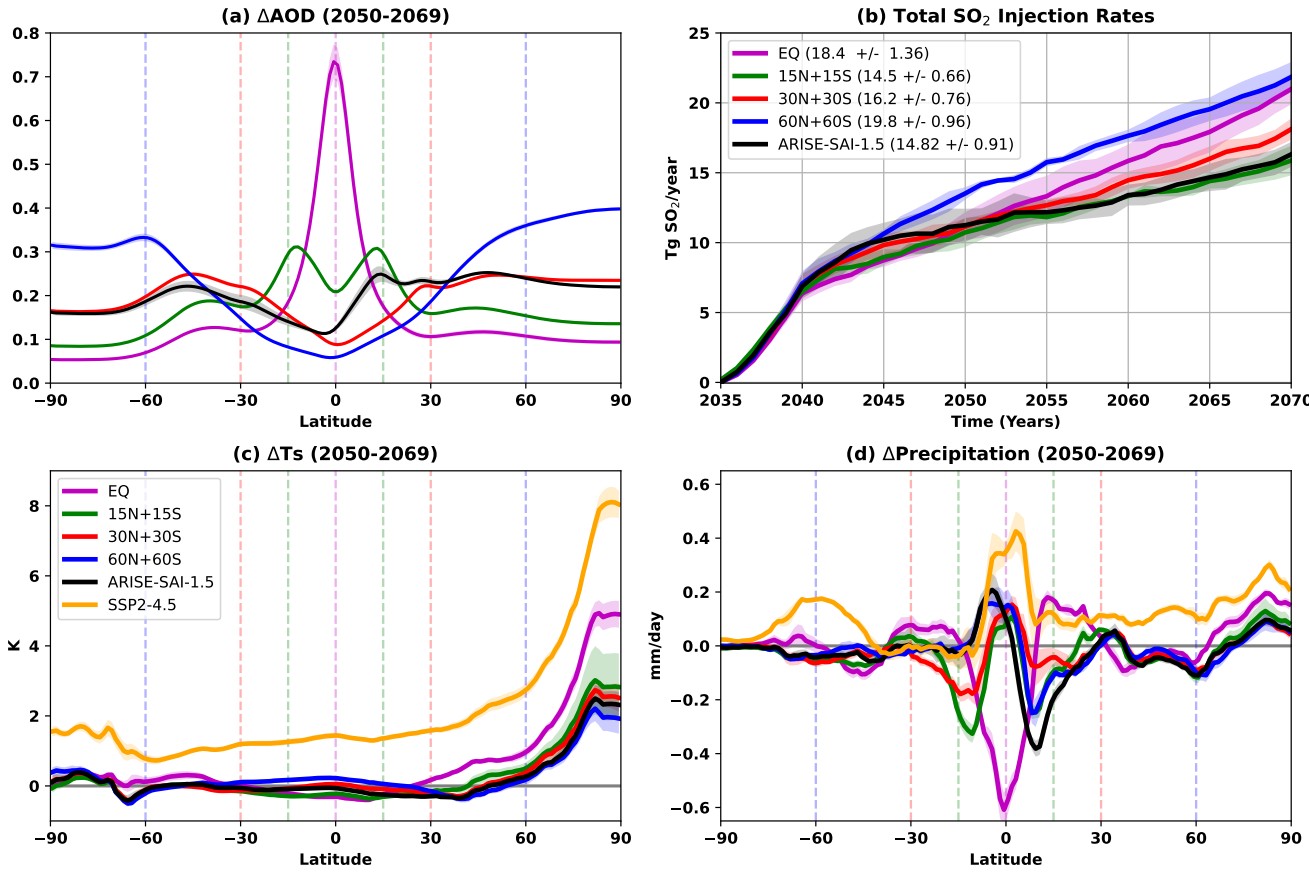

**Figure 2.** (a) The ensemble-mean change in aerosol optical depth (550nm, stratosphere only) in 2050-69 relative to SSP2-4.5. (Note that the baseline stratospheric AOD under SSP2-4.5 is three orders of magnitude smaller than the changes under SAI.) (b) The ensemble-mean total $SO_2$ injection rate (with the average over 2060-69 shown in the legend). The ensemble-mean surface air temperature (c) and precipitation (d) change in 2050-69 relative to the reference period (2014-33) for all simulation sets. The dashed lines in panels a, c, and d give the latitudes of injection. In all panels, the shading shows +/- one standard deviation of the ensemble.

The total $SO_2$ injection rate (panel b) shows that the most efficient injection strategies in UKESM1 in terms of cooling per Tg $SO_2$ are 15N+15S, 30N+30S and ARISE-SAI-1.5, with around 15 to 16 Tg $SO_2$ / year required to reach the temperature target by the end of the simulation (1.6K cooling averaged over 2050-69). Such injection magnitudes are comparable to a Pinatubo eruption which is estimated to have emitted between 14 and 23 Tg $SO_2$ (Guo et al., 2004). The relative injection amounts at each latitude in UKESM1 broadly agree with the CESM2 results in Zhang et al. (2024). The average lifetime of the injected stratospheric aerosols is 0.90 +/- 0.019, 0.87 +/- 0.024, 0.73 +/- 0.0094, 0.59 +/- 0.011, 0.80 +/- 0.020 years for the EQ, 15N+15S, 30N+30S, 60N+60S, and ARISE-SAI-1.5 simulations respectively. Here, the stratospheric aerosol lifetime (yr) is calculated as the ratio of the anomalous stratospheric $SO_2$ burden (Tg) to the injection rate (Tg/yr), averaged over the last 10 years of the simulations.

For the equatorial injection, the confinement of aerosols inside the tropical pipe leads to a very high AOD increase at the equator and a small increase outside the tropics compared to the other injection strategies. The larger injection rate is thus due to the lower efficacy of tropical forcing (Kang and Xie, 2014) and to the confinement of aerosols inside the tropical pipe, enhancing the formation of larger aerosols which sediment faster. Figure A1 shows the aerosol effective radius as calculated in Visioni et al. (2023a) (their equation 3) for one ensemble member of each set of SAI simulations, and confirms that the EQ simulations have much larger aerosols. The larger injection rates for 60N+60S, on the other hand, arise due to faster removal of aerosols when injected near the descending branch of the Brewer-Dobson Circulation, as evidenced by the shortest lifetime of stratospheric aerosols for 60N+60S. In addition, the scarcity of sunlight at high latitudes during parts of the year further reduces the overall cooling efficiency of the 60N+60S injection strategy. We note that this shortcoming can be overcome by injecting aerosols only in spring in each hemisphere (Lee et al., 2021). Zhang et al. (2024) also reported a larger injection amount needed for their 60N+60S simulations, although their CESM2 simulation injected $SO_2$ only in spring and at a lower altitude (15 km) than in UKESM1.

The zonal-mean annual-mean surface air temperature and precipitation changes relative to the target period (2014-33 of the SSP2-4.5 simulation ensemble-mean) are shown in panels c and d. While off-equatorial strategies manage to reduce the latitudinal temperature residuals between 30°N and 60°N to near zero, the EQ strategy has almost 1K of residual warming in that same region. In general, the zonal mean surface air temperature change does not differ by more than 1K between all SAI strategies, apart from North of 80°N where the EQ strategy undercools the Arctic leaving 4.4K of residual Arctic warming compared to 2.1K for the 60N+60S strategy. Remarkably, the 30N+30S and ARISE strategies only have 2.6K and 2.4K of residual Arctic warming respectively; hence they have a similar temperature change pattern to the 60N+60S strategy despite having a very different AOD pattern. This shows that no pattern of AOD from SAI is able to entirely cancel out the spatial forcing from greenhouse gases in the model. This is especially the case in the Arctic where greenhouse gases exert a longwave forcing year-round whereas no SAI aerosol shortwave forcing will occur during the polar winter. This mismatch in forcings is amplified by UKESM1's climate feedbacks, which have been noted to lead to a strong Arctic amplification in comparison to other models (Swaminathan et al., 2022), yielding a relatively strong residual Arctic warming for all AOD forcing patterns. We note that the large Arctic temperature change hides the pattern of surface temperature change elsewhere in figure 2c, hence figure A2(a) shows the temperature change excluding the Arctic region. Additionally, the smaller amount of cooling for the EQ

strategy (fig 1) may exaggerate the undercooling of the Arctic. Therefore figure A2(b) shows the surface temperature change for the SAI strategies relative to SSP2-4.5 normalized by the global-mean temperature change.

Finally, the zonal-mean precipitation in SSP2-4.5 increases everywhere except the Southern Hemisphere subtropics, and generally increases more where climatological precipitation is higher. For the EQ strategy, there is a significant reduction in precipitation at the equator (where climatological precipitation is high) and increase in precipitation in the subtropics (where climatological precipitation is low). This is consistent with a marked reduction in the Hadley Circulation intensity (figure A3) and with findings from Wells et al. (2024) (their figure 7). This pattern of precipitation change likely results from the SAI-induced tropical lower stratospheric heating (figure A4) (Simpson et al., 2019) as well as the reduction in the surface solar irradiance and associated reductions in latent and sensible heat fluxes, both of which are particularly evident in the tropics under the EQ strategy in UKESM1 owing to the high tropical sulfate and AOD (fig 2a; Visioni et al. (2023a); Wells et al. (2024)). For the 15N+15S strategy, precipitation decreases significantly near the injection latitudes. In general, unlike for surface air temperature changes, there are more marked differences in precipitation changes between the different strategies, which are explored further below. Figure A5 shows the zonal-mean change in surface air temperature and precipitation over land only for the ensemble-mean of SSP2-4.5 and all SAI strategies. The surface air temperature change patterns are broadly similar, though the strength of Arctic amplification is less accentuated over land. The increase in precipitation in SSP2-4.5 is muted over land and the decrease in precipitation at the equator over land is much larger (up to 1mm/day). The change in precipitation over land is otherwise broadly similar. We also show maps of the annual-mean ensemble-mean surface air temperature and precipitation changes in figures A6 and A7 respectively.

Both tropospheric and stratospheric aerosols are well known to have impacts on the position of the intertropical convergence zone (ITCZ). Figure 3 shows the latitude of the ITCZ in 2050-69 for each ensemble-mean along with the standard deviation of the 2050-69 mean of ensemble members (3 to 5 members depending on the ensemble), as a function of the interhemispheric surface temperature gradient $T1$ as defined in Kravitz et al. (2017) (their equation 1). The grey shading shows the standard deviation of the SSP2-4.5 ensemble in the target period (2014-2033). Here, the ITCZ is computed as the linear interpolation of the latitude near the equator where the zonal-mean mass streamfunction at 500 hPa changes sign. As discussed in Byrne et al. (2018), the ITCZ location is determined by the net energy input into the tropical atmosphere, which is affected by cloud and radiation processes, as well as ocean heat uptake. Their equation 5 shows that under a warmer world, the ITCZ will tend to shift towards the equator.

In the EQ strategy, the ITCZ shifts northward relative to the reference period. In the 15N+15S and 30N+30S strategies, the ITCZ is within the reference period's range. In the ARISE-SAI-1.5 strategy however, the ITCZ shifts southward by approximately 1.3 degrees; this is consistent with higher $SO_2$ injection rates in the Northern Hemisphere and the resulting higher tropical AOD in that hemisphere (figure 2a, Henry et al. (2023)). In the 60N+60S strategy, the ITCZ also shifts southward (by approximately 1.1 degrees); again there are asymmetries in the corresponding tropical AOD in that strategy, with slightly higher AOD in the northern hemisphere than the southern hemisphere. While these tropical AOD changes are much smaller than those in the EQ strategy, they also influence temperature gradients close to the equator. While Haywood et al. (2013, 2016) showed that preferential injection of stratospheric aerosols into the northern hemisphere leads to a southward shift in the ITCZ

in HadGEM2-ES, the predecessor of UKESM1, the more nuanced approach of Hawcroft et al. (2017) showed that it is more subtle changes in cross-equator temperature gradients that primarily influence the ITCZ position.

It is common in the stratospheric aerosol injection literature to control for T1 as a way of reducing changes in the location of the ITCZ. In ARISE-SAI-1.5, T1 is used by the controller to assess the interhemispheric temperature difference and minimise changes in the ITCZ location. For the SAI simulations, there is a correlation between the latitude of the ITCZ and the hemispheric difference in temperature (T1) as shown in figure 3 (dashed line), which is estimated here by fitting a line which minimises the least squared error. The linear function is as follows: ITCZ latitude = 11.8*T1 - 2.9. Based on this linear assumption, the predicted ITCZ latitude for the value of T1 simulated in the SSP2-4.5 ensemble mean in 2050-69 should be approximately 7°N. Its actual latitude (5.3°N) is found at lower latitude than the predicted one, suggesting that the warming itself may have shifted the ITCZ towards the equator as discussed above. We further note that a similar relationship inferred from the UKESM1 historical simulation suggests that a 0.4K change in T1 is needed to induce a 0.8 change in ITCZ latitude (fig A8). (This was not included in figure 3 for clarity purposes.) As such values also do not fit the relationship inferred from the SAI simulations above, the results highlight that factors other than the inter-hemispheric temperature gradient alone are important in modulating the ITCZ position. Thus, further developments of the controller might benefit from utilising more sophisticated metrics than a simple measure of interhemispheric temperature gradient to refine injection strategies, as has been demonstrated in Lee et al. (2020).

## 3.2 Stratospheric impacts

Figure 4 shows changes in zonal-mean zonal wind for each SAI strategy in 2050-69 relative to SSP2-4.5 in the same period (i.e. 2050-69), along with the locations of injection marked by black diamonds. The stratospheric jets are strengthened in all strategies except 60N+60S, with the strongest response for the equatorial injection. This is consistent with Bednarz et al. (2023a), and is caused by the anomalous increase in the equator-to-pole temperature gradient in the stratosphere as the result of aerosol-induced tropical lower stratospheric heating (figures A4 and A9) altering stratospheric winds via the thermal wind relationship and feedbacks with wave propagation and breaking. Since the strength of all these effects is roughly proportional to the magnitude of the aerosol-induced tropical lower stratospheric heating (figures A4 and A9), this explains the strong dependence of the magnitude of stratospheric vortex strengthening on the latitude of the injection. In the troposphere, all SAI strategies simulate the largest cooling in the tropical upper troposphere (Figure A9); this causes a year-round weakening of the subtropical jets, again with the largest changes for the equatorial injection. In the extratropics, stratospheric westerly responses can at certain times propagate down to the troposphere below in the form of a poleward shift of the eddy-driven jet (e.g. Bednarz et al. (2023a)); a suggestion of such a response is for instance found in the southern hemisphere summer under the equatorial injection (not shown).

The age of air refers to the transport time of air from the troposphere to the stratosphere and acts as a proxy for understanding stratospheric circulation, transport, and mixing. While it cannot be measured directly, it can be inferred from stratospheric measurements of conserved gases, such as carbon dioxide or sulfur hexafluoride ($SF_6$) (Waugh, 2009). Figure 5 shows the change in age of air, as output by the model, for each SAI strategy relative to SSP2-4.5 in 2050-69. In the EQ and 15N+15S

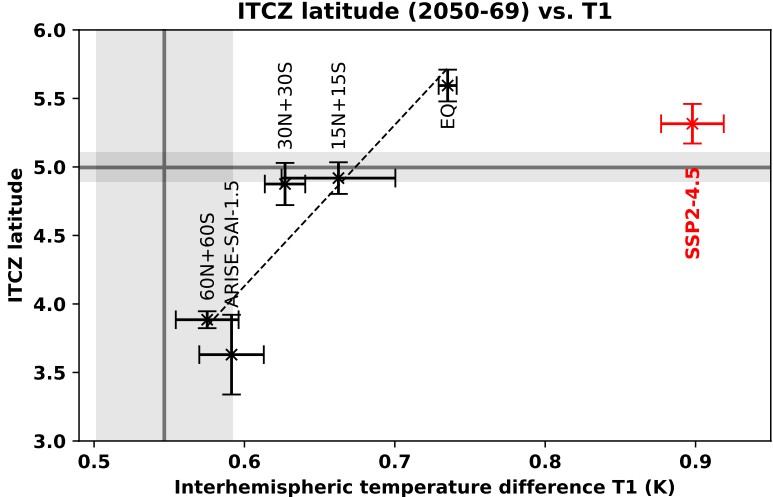

**Figure 3.** Latitude of the intertropical convergence zone (ITCZ) of the ensemble-mean of the different SAI strategies (black) and SSP2-4.5 (red) in 2050-69. The whiskers denote the standard deviation of the 2050-69 mean of ensemble members (3 to 5 members depending on the ensemble). The horizontal grey line is the mean ITCZ latitude of SSP2-4.5 in the reference period (2014-33). The x-axis is the interhemispheric temperature difference (T1) as defined by Kravitz et al. (2017) (their equation 1). The vertical grey line is the interhemispheric temperature difference (T1) in the SSP2-4.5 reference period (2014-33). The grey boxes show the standard deviation of the SSP2-4.5 ensemble in the reference period. The dashed line is a linear fit for the SAI simulations only, and the linear function is as follows: ITCZ latitude = 11.8*T1 - 2.9.

strategies, we find relatively older air in the upper troposphere and lower stratosphere (UTLS) region, which shows that the tropical upwelling in UTLS and the shallow branch of the Brewer-Dobson circulation (BDC) slow down as a result of SAI. We also find relatively younger air in the middle and upper stratosphere under these two SAI strategies, showing the associated acceleration of the deep branch of the BDC above the aerosol layer. Both of these effects are much weaker for injections away

from the tropics, in agreement with the smaller SAI-induced lower stratospheric heating (figure A4) and the resulting changes in planetary wave propagation and breaking (not shown, see e.g. Tilmes et al. (2018a); Bednarz et al. (2023a)). These SAI-induced changes in stratospheric circulation and transport modulate concentrations of stratospheric species, including ozone and sulfate aerosols, as well as the removal of aerosols from the stratosphere.

    Figure 6 shows the ensemble-mean change in ozone for each SAI strategy in 2050-69 as a percentage change relative to

SSP2-4.5 in the same period (2050-69), along with the location of injections marked by black diamonds. Also shown in panel d is the zonal-mean change in column ozone in 2050-69 relative to SSP2-4.5 in 2050-69. Comparing the same time period with and without SAI enables us to see a clearer picture as the signal from the long-term decline in ozone depleting substances and increase in greenhouse gases is removed, thus isolating the impact of the different SAI strategies. For reference, figure A10 shows the zonal-mean change in column ozone in 2050-69 for all simulation ensembles relative to the reference period (2014-

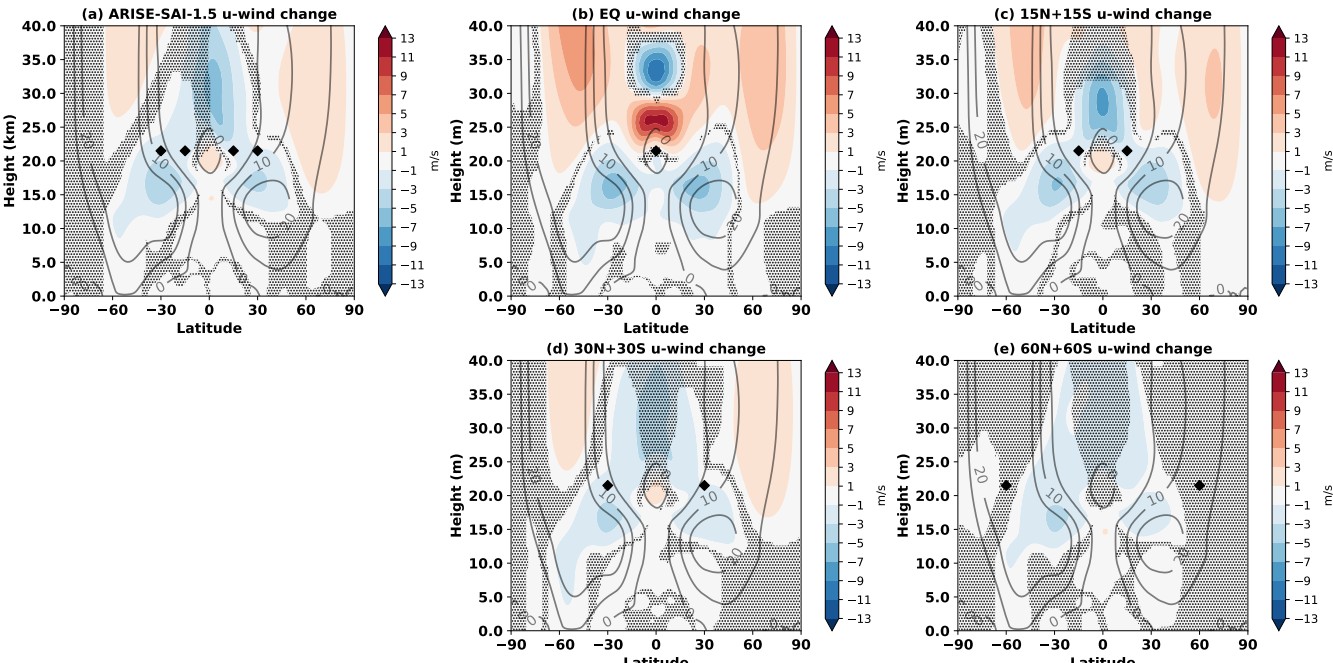

**Figure 4.** The ensemble-mean change in zonal wind for each SAI strategy in 2050-69 relative to SSP2-4.5 in 2050-69. The grey contour lines denote the horizontal wind values for the ensemble-mean of SSP2-4.5 in 2050-69 in m/s. The black diamonds give the location of injection. Gray shading indicates areas where the difference is not statistically significant, as evaluated using a double-sided t test with p<0.05 adjusted for the false discovery rate and considering all ensemble members and 20 years as independent samples.

33) and a time-series of the total ozone for all simulation ensembles, which increases as the concentration of ozone-depleting substances is reduced.

With the exception of the equatorial strategy, the annual mean total column ozone changes at different latitudes are relatively small (< 5 DU, fig 6d). More interesting structure is found when considering latitudinal cross-sections - in this case, the clearest common signal across all strategies is the increase in the tropical lower stratospheric ozone. This arises due to the aerosol-induced reduction in upwelling in the UTLS (as illustrated by older age-of-air in the lower stratosphere in fig 5) and the resulting reduction in the input of ozone-poor tropospheric air into the stratosphere (e.g Tilmes et al., 2018a; Bednarz et al., 2023a). This effect is strongest for the equatorial injections as it has the largest concentration of stratospheric aerosols in the tropics (fig 2a) and thus largest increases in tropical stratospheric temperature (fig A4).

In the EQ and the 15N+15S strategies, there is a ∼20% and ∼5% reduction, respectively, in stratospheric ozone just above the location of injection, with these changes dominating the corresponding total column ozone changes near the latitudes of the injection. This results from the acceleration of upwelling above the aerosol layer as the result of the aerosol-induced lower stratospheric warming and the subsequent impacts on stratospheric winds and wave propagation and breaking (Bednarz et al., 2023a). This increase in upwelling brings more ozone-poor air from the lower to mid-stratosphere, leading to local decreases

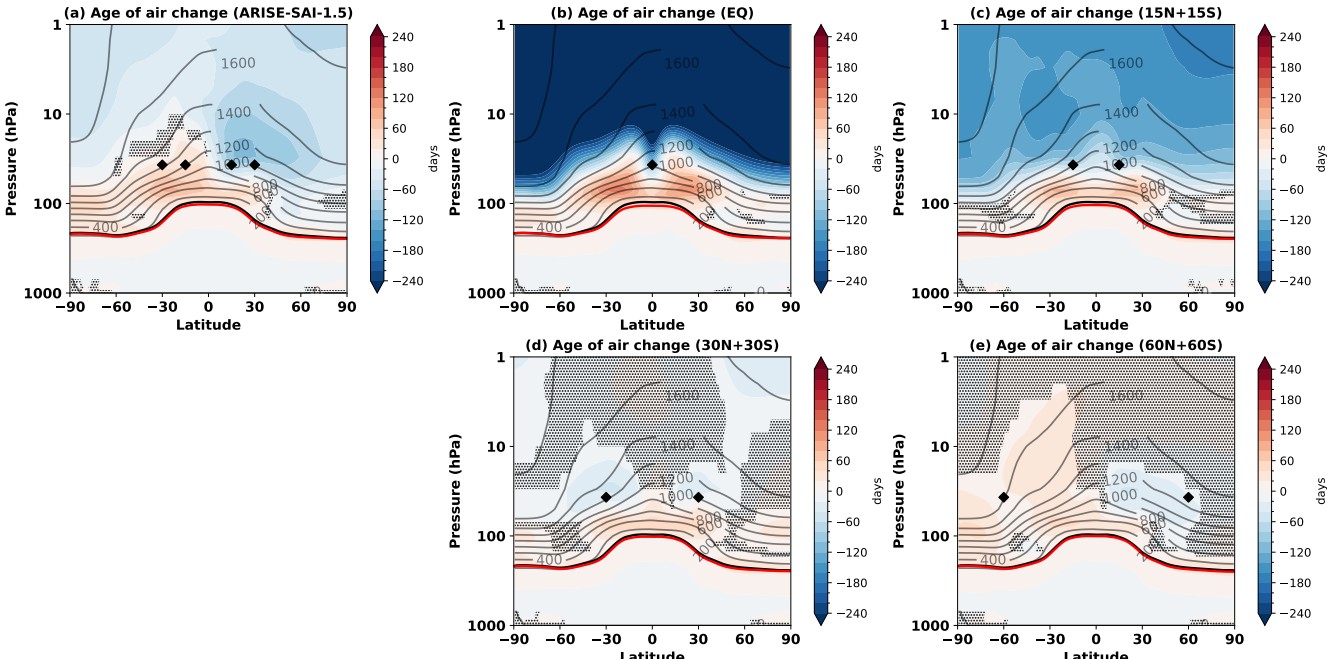

**Figure 5.** The ensemble-mean change in age of air for each simulation set in 2050-69 relative to SSP2-4.5 in 2050-69. The grey contour lines denote the age of air for the ensemble-mean of SSP2-4.5 in 2050-69 in days. The black line shows the tropopause in the SSP2-4.5 in 2050-69, and the red line is the tropopause in the SAI simulation in 2050-69. The black diamonds give the location of injection. Shaded areas indicate where the difference is not statistically significant, as evaluated using a double-sided t test with p<0.05 adjusted for the false discovery rate and considering all ensemble members and 20 years as independent samples.

in ozone in fig 6, as well as reduces mean age-of-air in most of the mid and upper stratosphere (fig 5). These changes are
consistent with Wells et al. (2024) (their figure 9).

Furthermore, there is a reduction in ozone in the extratropical stratosphere for the EQ strategy, likely caused by the aerosol-induced strengthening of the stratospheric polar vortices (see fig 4). These stronger and colder stratospheric polar vortices reduce mixing-in of ozone-rich midlatitude air into the polar stratosphere and colder temperatures enhance chemical ozone depletion (Rex et al., 2004; Tegtmeier et al., 2008; Bednarz et al., 2016). In the upper stratosphere, the equatorial strategy also
shows small but statistically significant ozone reductions at all latitudes, likely as the result of the enhanced $HO_X$-mediated ozone loss under aerosol-induced stratospheric moistening (Tilmes et al., 2018a, 2022), which is also largest in the EQ strategy due to the largest associated changes in tropical cold point tropopause temperatures.

While the 30N+30S and 60N+60S simulations do not lead to substantial changes in circulation (fig 4 and 5) and hence dynamically driven ozone changes, one would expect chemical ozone losses resulting from in-situ heterogeneous halogen re-
actions on aerosol surfaces to dominate the ozone response in the extratropical lower stratosphere, particularly in the Antarctic. However, we find no significant ozone reductions in these regions in the 30N+30S and 60N+60S simulations. This likely occurs

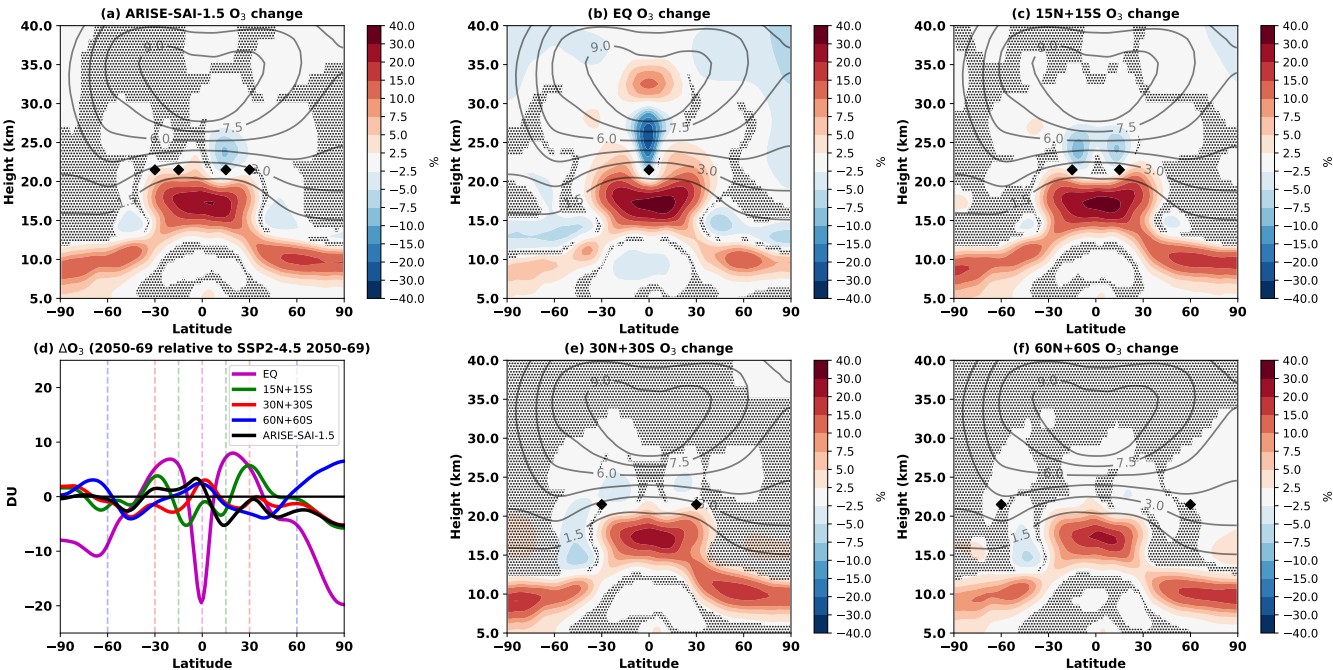

**Figure 6.** (a,b,c,e,f) The ensemble-mean change in ozone mixing ratios for each SAI strategy in 2050-69 as a percentage change relative to SSP2-4.5 in 2050-69. The grey contour lines denote the volume mixing ratio of ozone in ppm for the ensemble-mean of SSP2-4.5 in 2050-69. The black diamonds give the location of injection. Gray shading indicates areas where the difference is not statistically significant, as evaluated using a double-sided t test with p<0.05 adjusted for the false discovery rate and considering all ensemble members and 20 years as independent samples. (d) The change in zonal-mean column ozone for each SAI strategy in 2050-69 relative to SSP2-4.5.

as the most important heterogeneous halogen reaction ($HCl + ClONO_2$) is not included on sulphate aerosols in this version of UKESM1 (Dennison et al., 2019).

Finally, aside from enhancing halogen activation, sulphate aerosols facilitate the $N_2O_5$ hydrolysis reaction on their surfaces

($N_2O_5 + H_2O \rightarrow 2*HNO_3$) which acts to reduce active nitrogen concentrations and, thus, increase ozone in the middle strato-sphere. While this has an important effect in CESM2 (Tilmes et al., 2018b; Bednarz et al., 2023a), it does not have a large impact in UKESM1 despite this reaction occurring on sulphate surfaces, thus underlining the uncertainties in these processes and their parametrizations.

## 4   Conclusions

In this study we have compared five different sets of UKESM1 simulations of stratospheric aerosol injection (SAI) strategies using $SO_2$. The background simulation, global-mean temperature target, altitude, and season of injection are the same in all five sets of simulations in order to isolate the role of the latitude of injection. The background emission scenario is the CMIP6 SSP2-

4.5 scenario and the global-mean temperature target under SAI is 1.5 degrees above model preindustrial temperatures, which corresponds to the mean of 2014-33 in UKESM1. This is the first such comparison between different latitudes of injection for this scenario in UKESM1. It is inspired from a similar study using CESM2 (Zhang et al., 2024) with the only differences being that the high-latitude injections in CESM2 were done at a lower altitude and only in the spring of each hemisphere. It is important to analyse the impacts of different injection latitudes for SAI in a different Earth System Model to evaluate the robustness of the conclusions drawn from Zhang et al. (2024). In this study, one set of simulations injects at the equator (EQ), three sets of simulations use pairs of latitudes (15N+15S, 30N+30S, 60N+60S) and inject equal amounts of aerosols in each hemisphere, and one set injects at the combination of 15°N, 15°S, 30°N, and 30°S, adjusting the injection amount yearly at each location in order to not only satisfy the global-mean temperature target, but also the interhemispheric and equator-to-pole temperature targets. The next proposed set of Geoengineering Model Intercomparison Project simulations will consist of $SO_2$ injection at both 30°N and S and controlling only the global-mean temperature (Visioni et al., 2023b). Hence it is important to assess the merits of such a strategy relative to other choices in the latitude of injection and the number of objectives (thus the complexity of the control algorithm).

The main takeaways are that the 30N+30S strategy is the second most efficient strategy among those presented in this manuscript in terms of amounts of $SO_2$ needed (12% more injection than the most efficient 15N+15S strategy), and is among the strategies which have the smallest changes in precipitation, position of intertropical convergence zone, ozone concentrations and atmospheric circulation (both in the troposphere and stratosphere), which is broadly consistent with previous results using CESM2. In both observed trends and future projections, the Arctic warms much faster than the rest of the planet. The 30N+30S strategy leads to 5.1 K Arctic cooling compared to 5.6 K for the 60N+60S strategy, despite having a much more latitudinally homogeneous AOD distribution. This is different to CESM2 which has more than 1K extra Arctic cooling for their polar strategy, though the injection takes place at a lower altitude relative to UKESM1 and in the spring of each hemisphere. It is also worth noting that Arctic amplification is much less pronounced in CESM2 relative to UKESM1. While the 30N+30S strategy leads to around 1.9 K of tropical lower stratospheric warming compared to 1.1 K for 60N+60S, which results in larger consequences on atmospheric circulation and chemistry, these are still much smaller than for the equatorial and 15N+15S strategies (4.1 K and 3.3 K respectively). The strategy using three different temperature objectives (ARISE-SAI-1.5) has a larger ITCZ shift relative to 30N+30S, but otherwise presents similar outcomes for other metrics. This shows that controlling for the interhemispheric temperature difference might be insufficient to maintain the ITCZ latitude as it is influenced by a number of other factors. Future implementations of the controller might thus benefit from using better proxies for maintaining the ITCZ position as was done in Lee et al. (2020). The 60N+60S strategy also shows a significant southward shift in the ITCZ compared to SSP2-4.5, but leads to no substantial strengthening of the stratospheric jets or changes in Brewer-Dobson circulation. The 60N+60S strategy requires 30% more injection, though it is worth noting that injecting in the spring of each hemisphere may lead to better efficiencies at high latitudes (Lee et al., 2021), and may plausibly make the 60N+60S strategy more efficient than the 30N+30S strategy in UKESM1. Furthermore, our simulations do not account for any delivery limitations of current technologies. It might be argued that emissions into the stratosphere at significantly lower altitudes might be achievable with relatively few modifications to the current aircraft fleet at latitudes of 60N+60S owing to the low altitude of the tropopause.

Finally, the equatorial strategy leads to trapping of the aerosols inside the tropical pipe, thus resulting in the largest impacts on atmospheric temperatures and circulation. To achieve the same temperature target, the strategy requires 14% more injection relative to 30N+30S, and results in large reductions in tropical precipitation and total column ozone in the tropics, a marked reduction in the Hadley Circulation intensity, and a large tropical lower stratospheric warming. These are all consistent with findings from Wells et al. (2024), which used UKESM1 but with a different background scenario and target climate. The decrease in efficiency for equatorial injection is subtly different from conclusions drawn from volcanic eruptions using an earlier version of the climate model (HadGEM2-ES, Jones et al. (2017)), where the greatest cooling impact was found to be for high-altitude equatorial eruptions. These differences may be due to the altitude of injection being 23-28km in Jones et al. (2017), which is above the altitude of injection for this work (21.5km).

The conclusions drawn from the UKESM1 model are broadly consistent with similar studies using the CESM2 model (Zhang et al., 2024; Bednarz et al., 2023a), in that the 30N+30S strategy yields similar climate outcomes to the more complicated multi-objective SAI simulations (ARISE-SAI-1.5) and is one of the most efficient in terms of $SO_2$ injected to achieve the same temperature target. Bednarz et al. (2023a) also found that moving the injection location further away from the equator reduces tropical lower stratospheric heating and its resulting dynamical effects. The changes in total column ozone in UKESM1 have a broadly similar structure but the amplitude of change is smaller than in CESM2; this could be because of the generally smaller magnitude of the associated aerosol-induced stratospheric heating in UKESM1 compared to CESM2 (Bednarz et al., 2023a) as well as incomplete representation of heterogeneous halogen reactions on sulphate aerosols in this version of UKESM1. The shifts in ITCZ, however, are less pronounced in CESM2 (Zhang et al., 2024) and are inconsistent with the UKESM1 simulations. Thus, understanding what controls shifts in ITCZ in SAI simulations deserves more enquiry. Finally, both models agree that equatorial injection has the largest stratospheric heating, and concurrent changes in atmospheric circulation and ozone.

In conclusion, this work supports the idea that injection at 30N+30S aiming for a single global-mean temperature target is an adequate choice for a multi-model comparison, combining improved outcomes compared to the previously used equatorial strategy whilst maintaining relative design simplicity, thus enabling a larger number of climate modelling centres to participate.

*Code and data availability.* The code to reproduce the figures is available at https://github.com/matthewjhenry/Henry24_latdep/. The data for the UKESM1 SSP2-4.5 simulations is available on the Earth System Grid Federation database. The data for the UKESM1 SAI simulations will be uploaded to Zenodo upon acceptance of the manuscript.

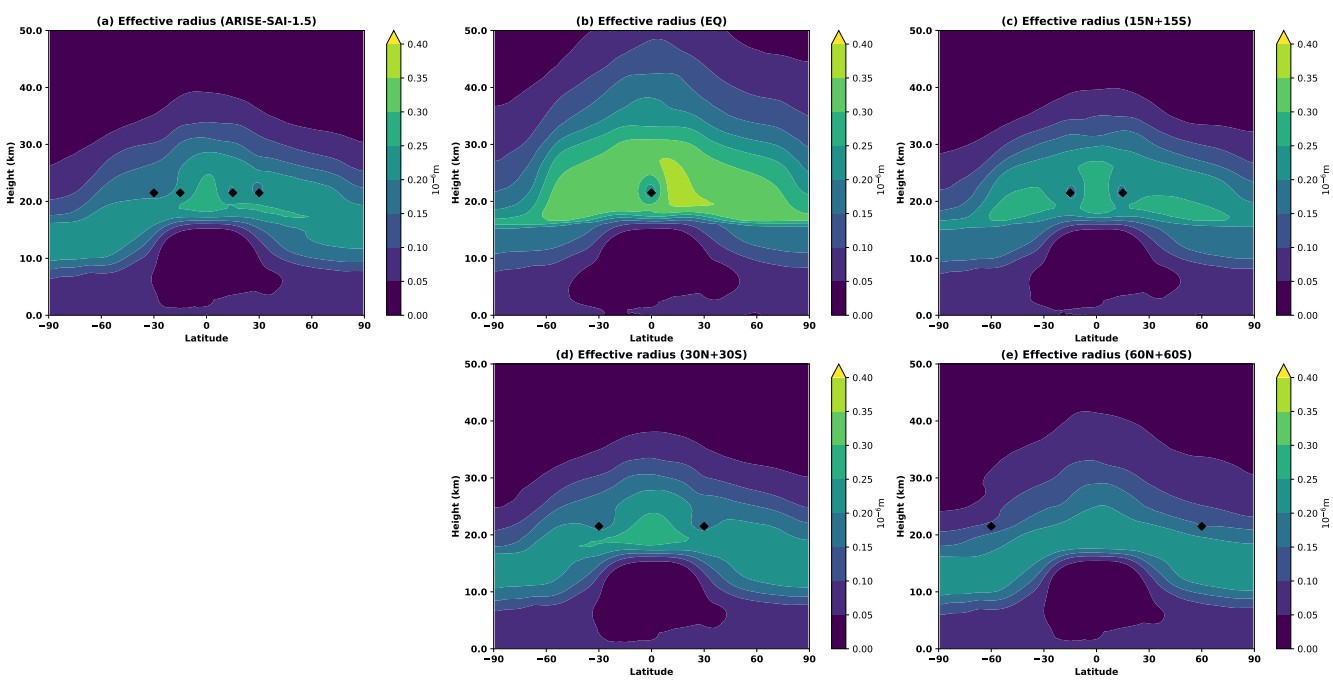

**Figure A1.** Effective radius for one ensemble member of the ARISE-SAI-1.5 (a), EQ (b), 15N+15S (c), 30N+30S (d), and 60N+60S (e) ensembles, as calculated in Visioni et al. (2023a) (their equation 3).

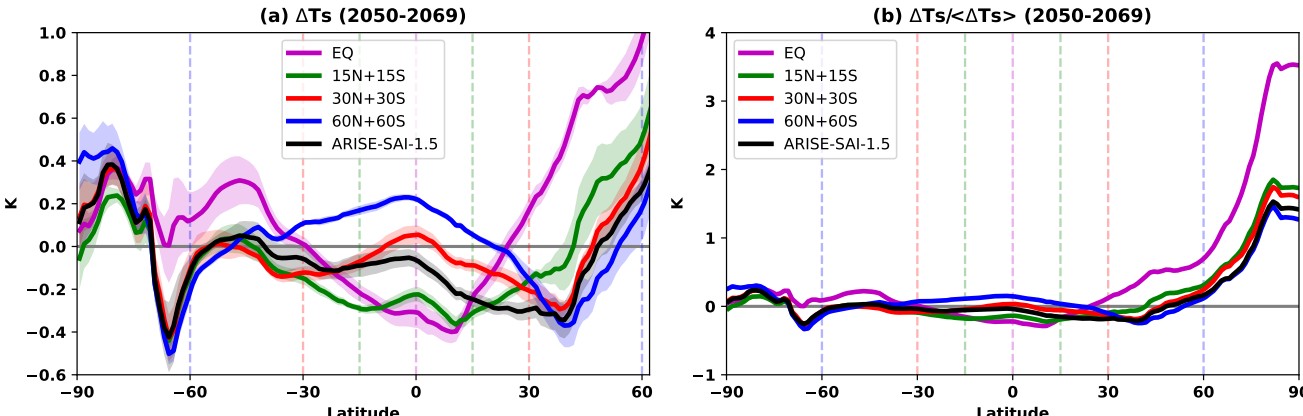

**Figure A2.** (a) Zonal-mean surface temperature change relative to SSP2-4.5 for the SAI strategies excluding the Arctic region (North of 60°N). The shading corresponds to +/- 1 standard deviation of the ensemble. (b) Zonal-mean surface temperature normalized by the global-mean surface temperature change (shown in the legend of figure 1).

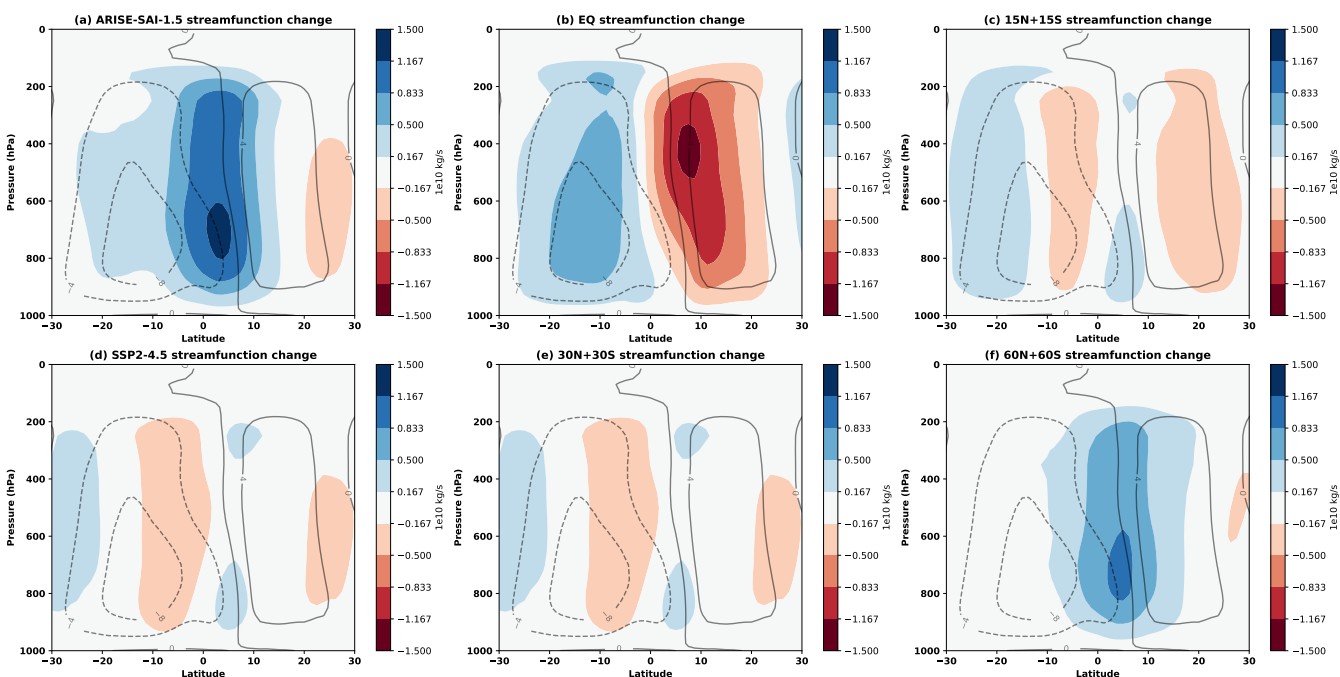

**Figure A3.** Change in annual-mean streamfunction for the ensemble-mean of each simulation set in 2050-69 relative to the reference period (2014-33). The black contour lines denote the streamfunction in the reference period. The blue contours and solid lines are associated with clockwise circulation, and the red contours and dashed lines are associated with anticlockwise circulation.

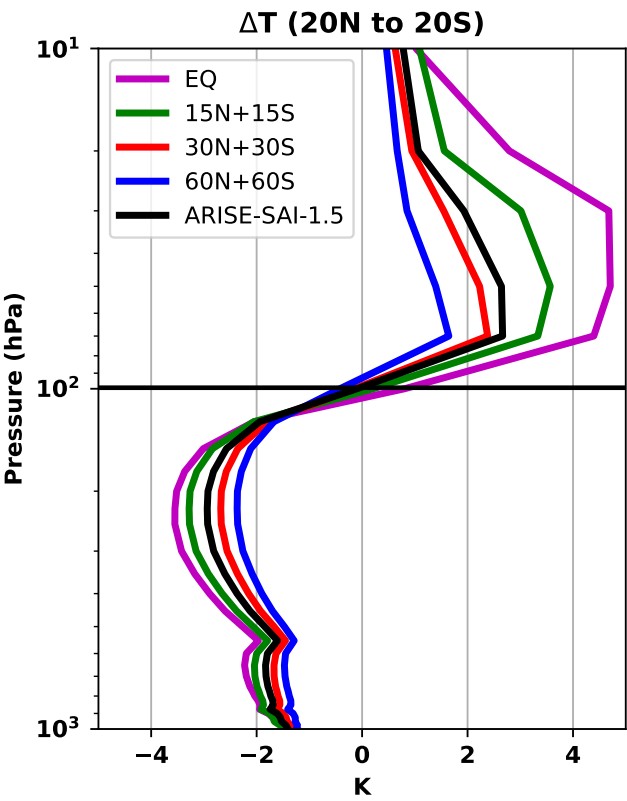

**Figure A4.** Difference in tropical (between 20°N and 20°S) atmospheric temperature between each SAI ensemble-mean and SSP2-4.5 in 2050-69.

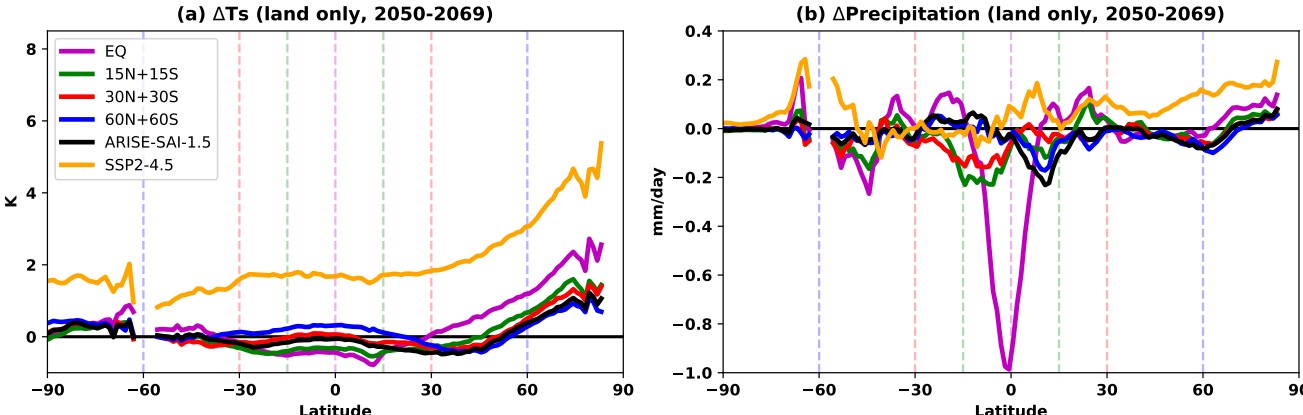

**Figure A5.** Same as figure 2c,d but over land only.

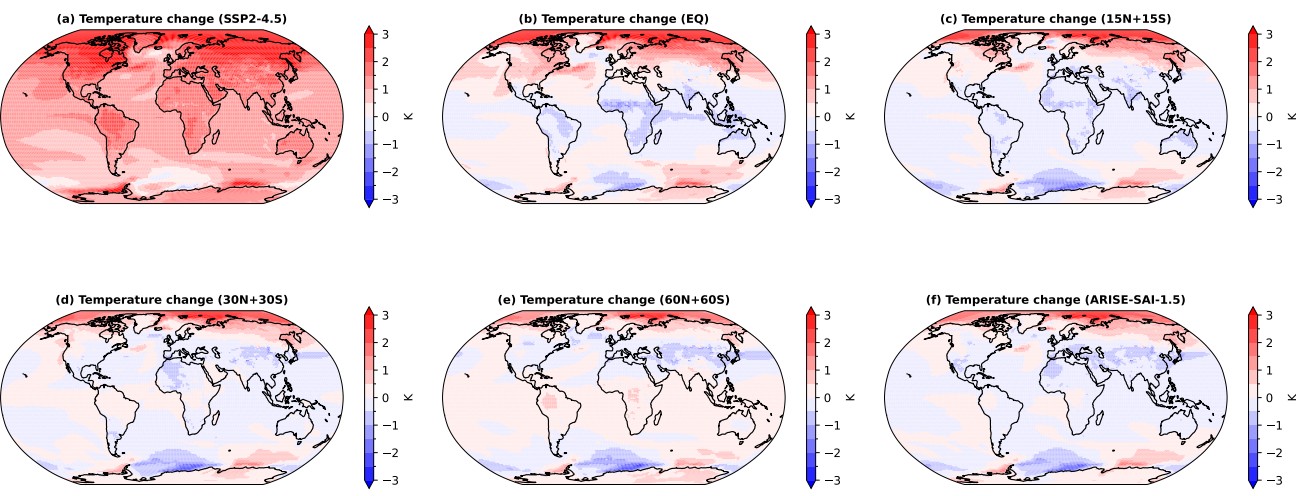

**Figure A6.** Maps of the ensemble-mean temperature change relative to the target period (2014-33) in SSP2-4.5.

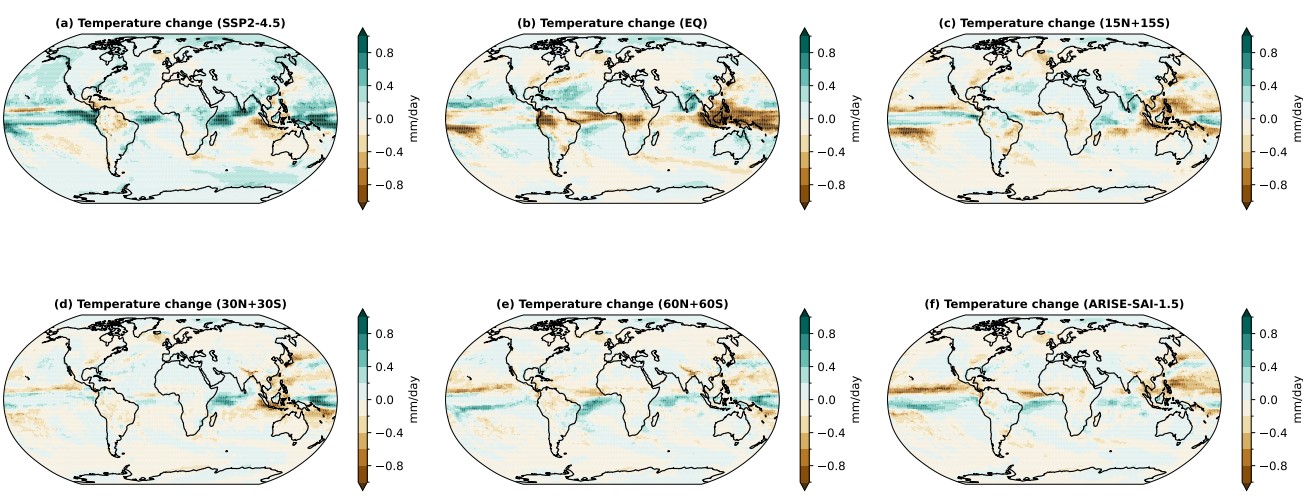

**Figure A7.** Maps of the ensemble-mean precipitation change relative to the target period (2014-33) in SSP2-4.5.

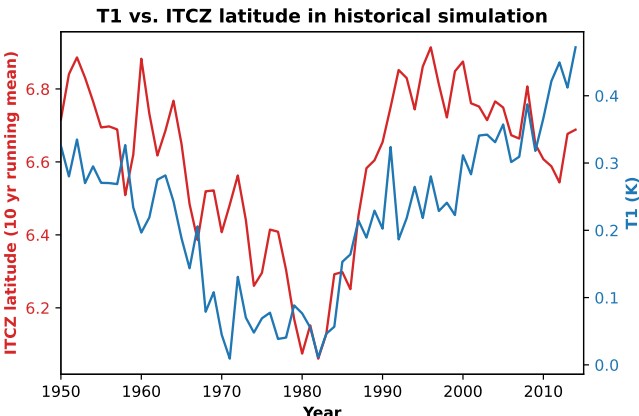

**Figure A8.** Relationship between T1 (a measure of interhemispheric temperature difference) and the 10-year running mean of the latitude of the InterTropical Convergence Zone for a UKESM1 simulation of the historical period (Sellar et al., 2019).

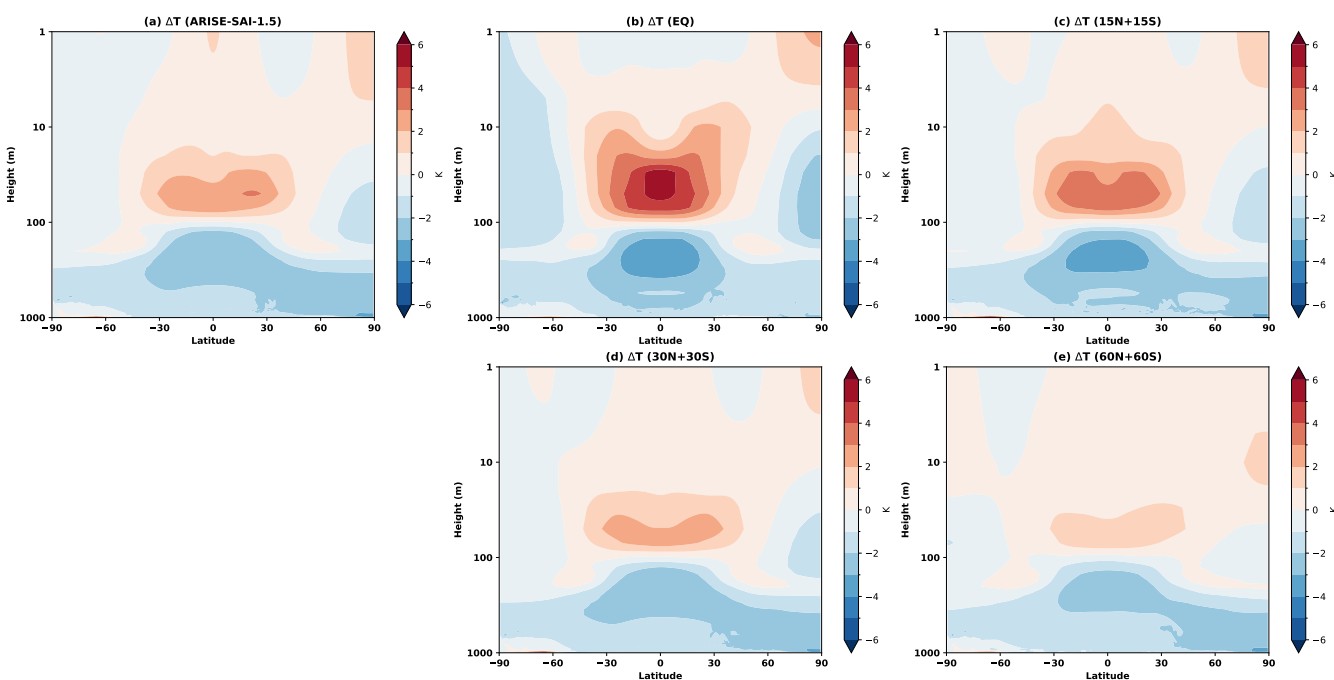

**Figure A9.** Atmospheric temperature difference between each SAI strategy ensemble-mean and SSP2-4.5 in 2050-69.

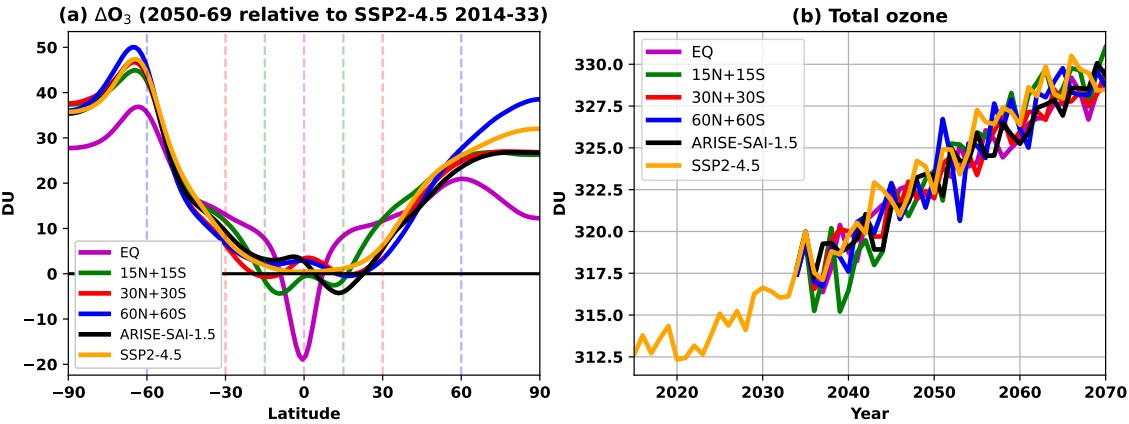

**Figure A10.** Difference in ozone in 2050-69 relative to SSP2-4.5 in the reference period (a), relative to SSP2-4.5 in 2050-69 (b), and the total ozone for all simulation ensemble-means (c).

*Author contributions.* MH wrote the manuscript and made the figures with contributions from all co-authors.

*Competing interests.* One of the authors is an editor for the Atmospheric Chemistry and Physics journal.

*Acknowledgements.* MH and JMH are funded by the Natural Environment Research Council Exeter-NCAR (EXTEND) collaborative development grant (NE/W003880/1) and by SilverLining through the Safe Climate Research Initiative. EMB acknowledges support from the NOAA cooperative agreement NA22OAR4320151 and the NOAA Earth's Radiative Budget program. The authors would like to thank Dou-
320 glas MacMartin for his feedback on the manuscript. The authors would additionally like to thank Douglas MacMartin and his colleagues at Cornell University for graciously providing their version of the controller and colleagues at the Met Office for implementation of this controller within UKESM1. For the purpose of open access, the author has applied a Creative Commons Attribution (CC BY) licence to any Author Accepted Manuscript version arising.

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
