# Peer review of "How Does the Latitude of Stratospheric Aerosol Injection Affect the Climate in UKESM1?"

_EGUsphere, 2024_

## Referee Comment (RC1)

**July 15th, 2024**

**Reviewer comment** regarding How Does the Latitude of Stratospheric Aerosol Injection Affect the Climate in UKESM1? submitted to Atmospheric Chemistry and Physics by Matthew Henry, Ewa M. Bednarz, and Jim Haywood

**General Comments**

This study examines five sets of simulations of different stratospheric aerosol injection (SAI) strategies carried out with the UK Earth System Model (UKESM1). The strategies are designed to be similar, with the primary difference being the latitude of sulfur injection; the strategies include equatorial injection; hemispherically symmetric injection at 15°N/S, 30°N/S, and 60°N/S; and the four-latitude injection strategy prescribed by the ARISE-SAI-1.5 protocol. The study considers the required sulfur injection rates to maintain a global mean temperature of PI + 1.5°C with each strategy, as well as some changes to surface and stratospheric climate.

Overall, the manuscript is sound, and I recommend minor revisions before it is formally published. My main comments are as follows:

- Many of the figures need more detail, either in the labeling, the data they present, or both. Many tend to show only ensemble means, zonal means, and 10-year averages; I would like to see more data from individual ensemble members represented, as well as more full maps and timeseries where appropriate. I would also like to see more information about the baseline/reference/target state to which the strategies are compared (see specific comments below).
- The authors assert in several places that the 30°N + 30°S strategy is the "best" or "optimal" strategy, and they claim that 30N+30S has the least overall "unwanted side effects," while equatorial injection has the most. While the authors are of course entitled to their opinion about which strategy is optimal or what constitutes a "negative" side effect (and I do not necessarily disagree!), such subjective assertions are unscientific; I would much rather see objective statements about the magnitude of disruptions to circulation or precipitation (the conclusion does a much better job of this).
- I disagree with the authors' decision to simulate year-round 60°N + 60°S injection at 22 km altitude. Past simulations of high-latitude injection with which I am familiar - Bednarz, et al.

(2023)[1]; Zhang, et al. (2024)[2]; Lee, et al. (2023)[3] - choose to inject around 15 km altitude (a short distance above the tropopause) rather than at the same altitude as subtropical injections; additionally, they choose to inject only in the polar spring, rather than in all months of the year. While I understand the authors' decision to prioritize consistency with the other strategies, consistency in the injection distance above the tropopause would have been more relevant than absolute altitude - I don't think an analysis of year-round 60°N + 60°S injection at 22 km is particularly meaningful, because at the current state of the research, this is not a strategy that would be considered either for inter-model comparison or in real life. Additionally, if one of the study's primary goals is to compare these strategies in UKESM and CESM, that cannot be done for the 60°N+60°S strategy if they were implemented so differently.

While I think the manuscript would suitable for publication without this change, I strongly encourage the authors to consider running at least one ensemble member of the 60°N + 60°S strategy with injection at a lower altitude (either 15 km, to be consistent with other studies, or the same distance above the tropopause in this model as the subtropical injection strategies), and ideally with injection only in the polar spring as well. I don't know what constraints the authors have on computational resources, or whether they are constrained by length limits for this journal, and this may not be feasible for them. However, I do not think the 60°N + 60°S strategy presented here will receive as much attention because there are no similar simulations (to my knowledge) by other models, and I do not expect there will be. If the authors are able to add what I consider to be a more realistic implementation of 60°N + 60°S injection, I think it would significantly increase the impact of this paper.

**Specific Comments**
- Lines 6-8, "many undesirable side effects" - I don't disagree, but I advise against using subjective words like "undesirable" and try instead for objective words (perhaps "disruptive"). Additionally, be careful to discriminate between the impacts of SAI, and the impacts of global warming which SAI does not prevent - for example, equatorial SAI "leading to" residual Arctic warming may be

[1] Bednarz, E. M., Butler, A. H., Visioni, D., Zhang, Y., Kravitz, B., and MacMartin, D. G.: Injection strategy – a driver of atmospheric circulation and ozone response to stratospheric aerosol geoengineering, Atmos. Chem. Phys., 23, 13665–13684, https://doi.org/10.5194/acp-23-13665-2023, 2023.

[2] Zhang, Y., MacMartin, D. G., Visioni, D., Bednarz, E. M., and Kravitz, B.: Hemispherically symmetric strategies for stratospheric aerosol injection, Earth Syst. Dynam., 15, 191–213, https://doi.org/10.5194/esd-15-191-2024, 2024.

[3] Lee, W. R., MacMartin, D. G., Visioni, D., Kravitz, B., Chen, Y., Moore, J. C., et al. (2023). High-latitude stratospheric aerosol injection to preserve the Arctic. *Earth's Future*, 11, e2022EF003052. https://doi.org/10.1029/2022EF003052

technically correct, but I would prefer to see wording such as "substantially undercools the Arctic".

- Lines 11-13, "we demonstrate that the 30N+30S strategy has, on balance, the least negative side effects" - this is a long way away from being shown. While this may be the authors' opinion concerning the five SAI strategies simulated with this model and the impacts considered in this paper, such a subjective blanket statement is unscientific; there are many, many more impacts to consider than just the ones examined here, and as the results do not show that the 30N+30S strategy is universally better than the others (for example, 15N+15S and ARISE-SAI-1.5 have smaller injection rates at the end of the experiment), and others may read this study and form a different opinion.

- Section 2 in general: please clarify which simulations are being presented here for the first time, and which were first presented in other studies - from what I can tell, the ARISE-SAI-1.5 and SSP2-4.5 simulations were first presented in Henry, et al. (2023)[4], and the other four SAI simulations are new here. As a whole, I think this section would be easier to interpret if you introduced SSP2-4.5 and ARISE-SAI-1.5, and the reference period and temperature targets, first, and then explained that you're presenting four new simulations which only target global mean temperature.

- Lines 69-70, "the global mean surface temperatures value in UKESM1 exceeds its preindustrial value by 1.5K" - this needs a citation, since the presented temperature data in Figure 1 only goes back to 2020 (I assume these numbers, and the temperature targets, were first presented in Henry, et al. 2023).

- Table 1: I would add a line to include SSP2-4.5, which is a unique set of simulations you are considering, and in line with my comment above, you could add a "first presented in" column with a citation/source for each simulation, even if it just says "Henry, et al. (2023)" for SSP2-4.5 and ARISE-SAI-1.5 and "here" for the rest. These changes would make it easier for a reader who is less familiar with the literature to follow along.

- Figure 1: the plot needs to be labeled better - clearly, the y-axis is relative to the preindustrial, but this is not stated. Additionally, the dashed black line is probably shown to represent the reference/target temperature, but it is not included in the legend.

- Figure 1: I would really like to see the individual ensemble members represented on this plot, not just the ensemble mean.

[4] Henry, M., Haywood, J., Jones, A., Dalvi, M., Wells, A., Visioni, D., Bednarz, E. M., MacMartin, D. G., Lee, W., and Tye, M. R.: Comparison of UKESM1 and CESM2 simulations using the same multi-target stratospheric aerosol injection strategy, Atmos. Chem. Phys., 23, 13369–13385, https://doi.org/10.5194/acp-23-13369-2023, 2023.

- Figure 1: I would also like to see actual numbers for global mean temperature somewhere, not just relative to the reference period - right now, the numbers are buried in the text; adding a right-side axis with the actual temperature values in °C or K, or adding the target value as text next to the dashed black line, would help give the reader another point of reference.

- Figure 1: the target and ARISE-SAI-1.5 both in black is a bit confusing - I would either pick a different color for one of them, or make the dashed line for the target thinner to more easily differentiate.

- Line 82 - saying Figure 2 "summarizes the climate response" is a bit strong… climate response is a lot more than zonal mean surface temperature and precipitation

- Line 85, "the most latitudinally homogeneous AOD is achieved by injecting at 30N+30S" - is it? The ARISE-SAI-1.5 AOD looks pretty similar to me. Additionally, you should clarify that this is out of the five strategies you considered here, and that this isn't a blanket truth for SAI

- Line 90, "the most efficient injection strategies" - 60N+60S isn't that different from the other three; certainly, the difference between 60N+60S and any of the other off-equatorial strategies is much larger than the difference between equatorial and off-equatorial

- Lines 93-94 - I disagree with this characterization as "optimal," for the same reasons as above

- Lines 99-100, "The larger injection rates for 60N+60S, on the other hand, arise due to faster removal of aerosols when injected near the descending branch of the Brewer Dobson Circulation (BDC)" - this needs a citation or other evidence to support it. This is one plausible contributing factor, but no analysis of aerosol lifetime is presented; polar SAI could simply cool the planet less efficiently because you're mainly inducing forcing changes over a relatively smaller fraction of the planet, and you're also injecting in winter when there's no sunlight to reflect

- Figure 2, all panels: again, I would really like to see individual ensemble members, not just ensemble mean

- Figure 2a: does "AOD" refer to 550nm AOD in the stratosphere only, or the whole atmosphere?

- Figure 2a: we need to see at least some information about the baseline SSP2-4.5 AOD here. I would plot raw AOD instead of $\Delta$AOD, and show the SSP2-4.5 in yellow as well; if SSP2-4.5 AOD is so small that it wouldn't even show up on this graph (e.g., order 0.01 or lower), you could just omit it entirely, and say in the text or figure caption that the baseline is several orders of magnitude smaller than the changes under SAI.

- Figure 2b: I would really like to see timeseries instead of 10-year averages. Additionally, there is no description of what the errorbars represent; however, I would rather just see individual ensemble members here too.

- Figures 2c and 2d: using a black line for both target and ARISE-SAI-1.5 is a bit confusing - please either change the color of one, or make the line for the target thinner to differentiate

- Figure 2d: while zonal mean surface temperature changes are probably okay, I think precipitation would be much better shown as a map. I know this would take up a lot of space, but you have maps for each strategy in figures 4-6; I don't know if you are constrained by length limits or number of figures, but it might be better to have one figure for AOD and injection rates (perhaps merged with Figure 1), and have a second figure containing temperature and precipitation. If this isn't feasible, I encourage you to add maps of precipitation changes to the supplementary.

- Lines 110-111, "no pattern of AOD from SAI is able to entirely offset the forcing from greenhouse gasses in the model" - firstly, this paragraph discusses temperature change, not forcing; secondly, this is only shown for the five patterns considered here

- Lines 133-134 - more detailed methodology is needed here; are you interpolating between grid boxes to compute this? Is this a linear interpolation, or a cubic one?

- Line 141 - I am not convinced this relationship is linear. You have fit a linear trend to the data, but that does not mean the relationship is linear

- Figure 3: how are you computing the dashed line? More details are needed here

- Figure 3: rather than using the bars to denote ensemble spread, I suggest just using different color dots for each of the individual simulations, as the number of individual simulations is relatively small

- Lines 156-158 - there is already at least some literature on this; see Lee, et al. (2020)[5]

- Figure 4 (and Figure 6) - height in km, or pressure in hPa, would probably be easier to read than height in m

- Line 171 - if age of air cannot be measured directly, can you elaborate on how it is presented here? Do you compute it from the model output, or is it computed and saved directly as model output during the simulation?

- Conclusion, in general: the results here are primarily compared to similar experiments in CESM2, but I am also curious about the robustness of these results in UKESM - for example, Wells, et al. (2024)[6] looked at equatorial injection vs. a four-latitude controller strategy similar to ARISE-SAI-1.5, but with a different background; if that study presented any of the same

[5] Lee, W., MacMartin, D., Visioni, D., and Kravitz, B.: Expanding the design space of stratospheric aerosol geoengineering to include precipitation-based objectives and explore trade-offs, Earth Syst. Dynam., 11, 1051–1072, https://doi.org/10.5194/esd-11-1051-2020, 2020.

[6] Wells, A. F., Henry, M., Bednarz, E. M., MacMartin, D. G., Jones, A., Dalvi, M., and Haywood, J. M.: Identifying climate impacts from different Stratospheric Aerosol Injection strategies in UKESM1, Earth's Future, 12, e2023EF004 358, 2024.

diagnostics as this study (ITCZ position, circulation changes, ozone depletion), it would be a useful point of comparison.

- Conclusion, in general: I like the objectivity of this section a lot more than the abstract/introduction; rather than saying that 30N+30S is the "best" strategy, evidence is presented that certain possibly unwanted impacts are smaller or absent. The equatorial strategy is still described as having "the most negative side effects" in lines 252 and 270, which I would advise rewording. Lastly, in line 272, I might choose a word such as "supports" instead of "confirms".

---

## Author Response (AR1)

**Response to reviewers for "How Does the Latitude of Stratospheric Aerosol Injection Affect the Climate in UKESM1?" by M. Henry, E.W. Bednarz, and J. Haywood**

In what follows the reviewer comments will be in black and our response will be in *red and italics*. We thank the reviewers for thorough and helpful comments, we think they have significantly ameliorated the manuscript.

Please note that there was a factor of 2 error in the amount injected at the Equator, which was a data processing issue. We didn't re-run any simulations and the error only affected fig 2b and some of the analysis. The relative injected amounts for each strategy are now much more in line with previous work done with CESM2.

**Reviewer 1**

General Comments

This study examines five sets of simulations of different stratospheric aerosol injection (SAI) strategies carried out with the UK Earth System Model (UKESM1). The strategies are designed to be similar, with the primary difference being the latitude of sulfur injection; the strategies include equatorial injection; hemispherically symmetric injection at 15°N/S, 30°N/S, and 60°N/S; and the four-latitude injection strategy prescribed by the ARISE-SAI-1.5 protocol. The study considers the required sulfur injection rates to maintain a global mean temperature of PI + 1.5°C with each strategy, as well as some changes to surface and stratospheric climate. Overall, the manuscript is sound, and I recommend minor revisions before it is formally published.

*We thank the reviewer for the positive review and detailed feedback.*

My main comments are as follows:
● Many of the figures need more detail, either in the labeling, the data they present, or both. Many tend to show only ensemble means, zonal means, and 10-year averages; I would like to see more data from individual ensemble members represented, as well as more full maps and timeseries where appropriate. I would also like to see more information about the baseline/reference/target state to which the strategies are compared (see specific comments below).

*We have significantly revised the figures. See more specific comments below.*

● The authors assert in several places that the 30°N + 30°S strategy is the "best" or "optimal" strategy, and they claim that 30N+30S has the least overall "unwanted side effects," while equatorial injection has the most. While the authors are of course entitled to their opinion about which strategy is optimal or what constitutes a "negative" side effect (and I do not necessarily disagree!), such subjective assertions are unscientific; I would much rather see objective

statements about the magnitude of disruptions to circulation or precipitation (the conclusion does a much better job of this).

*We agree and have removed the subjective language around 30N+30S and equatorial injection in the abstract and the main text.*

● I disagree with the authors' decision to simulate year-round 60°N + 60°S injection at 22 km altitude. Past simulations of high-latitude injection with which I am familiar - Bednarz, et al.(2023)[1]; Zhang, et al. (2024)[2]; Lee, et al. (2023)[3] - choose to inject around 15 km altitude (a short distance above the tropopause) rather than at the same altitude as subtropical injections; additionally, they choose to inject only in the polar spring, rather than in all months of the year. While I understand the authors' decision to prioritize consistency with the other strategies, consistency in the injection distance above the tropopause would have been more relevant than absolute altitude - I don't think an analysis of year-round 60°N + 60°S injection at 22 km is particularly meaningful, because at the current state of the research, this is not a strategy that would be considered either for inter-model comparison or in real life. Additionally, if one of the study's primary goals is to compare these strategies in UKESM and CESM, that cannot be done for the 60°N+60°S strategy if they were implemented so differently.

While I think the manuscript would suitable for publication without this change, I strongly encourage the authors to consider running at least one ensemble member of the 60°N + 60°S strategy with injection at a lower altitude (either 15 km, to be consistent with other studies, or the same distance above the tropopause in this model as the subtropical injection strategies), and ideally with injection only in the polar spring as well. I don't know what constraints the authors have on computational resources, or whether they are constrained by length limits for this journal, and this may not be feasible for them. However, I do not think the 60°N + 60°S strategy presented here will receive as much attention because there are no similar simulations (to my knowledge) by other models, and I do not expect there will be. If the authors are able to add what I consider to be a more realistic implementation of 60°N + 60°S injection, I think it would significantly increase the impact of this paper.

*That's a fair point and something we have considered as we were inspired by Zhang et al. However, the focus of this study is to isolate the effect of the latitude of injection, so we chose the same altitude and seasonality as the other simulations so as to not add some confounding factors. Note that the title is "How does the Latitude of Stratospheric Aerosol Injection Affect the Climate"; it's not supposed to address both latitude and altitude and we are quite clear about this in the title of the paper. Additionally, the lifetime of the aerosols is longer when injected at this higher altitude, thus the seasonality of injection will have a much smaller effect. We believe that even if the reviewer is correct and the 60N+60S strategy in our study is unlikely to be considered for a future inter-model comparison or real-life deployment, there is a merit in systematically exploring the role of injection latitude in order to better understand the underlying physical mechanisms and model uncertainties, thus aiding both any future strategy design efforts as well as impact studies. We agree with the reviewer, however, that this is an important aspect to pursue in future follow up studies, and we plan to work further with UKESM1 to focus*

*specifically at high-latitude low-altitude injection and the role of altitude and seasonality as suggested at the 14th Annual GeoMIP meeting that took place recently at Cornell, USA.*

*We add the following to the conclusions:*

*"Furthermore, our simulations do not account for any delivery limitations of current technologies. It might be argued that emissions into the stratosphere at significantly lower altitudes might be achievable with relatively few modifications to the current aircraft fleet at latitudes of 60N+60S owing to the low altitude of the tropopause."*

**Specific Comments**

● Lines 6-8, "many undesirable side effects" - I don't disagree, but I advise against using subjective words like "undesirable" and try instead for objective words (perhaps "disruptive"). Additionally, be careful to discriminate between the impacts of SAI, and the impacts of global warming which SAI does not prevent - for example, equatorial SAI "leading to" residual Arctic warming may be technically correct, but I would prefer to see wording such as "substantially undercools the Arctic".

*Agreed, we have edited the text as suggested.*

● Lines 11-13, "we demonstrate that the 30N+30S strategy has, on balance, the least negative side effects" - this is a long way away from being shown. While this may be the authors' opinion concerning the five SAI strategies simulated with this model and the impacts considered in this paper, such a subjective blanket statement is unscientific; there are many, many more impacts to consider than just the ones examined here, and as the results do not show that the 30N+30S strategy is universally better than the others (for example, 15N+15S and ARISE-SAI-1.5 have smaller injection rates at the end of the experiment), and others may read this study and form a different opinion.

*That's a fair point, we have moderated the last sentence of the abstract as follows: "Finally, while all the SAI strategies come with trade-offs, our work shows that the 30N+30S strategy is a good candidate strategy for an inter-model comparison, and is easier to implement than a multi-latitude controller algorithm which has been adopted by GeoMIP under the G6-SAI-1.5K strategy (Visioni et al., 2024)."*

● Section 2 in general: please clarify which simulations are being presented here for the first time, and which were first presented in other studies - from what I can tell, the ARISE-SAI-1.5 and SSP2-4.5 simulations were first presented in Henry, et al. (2023)4, and the other four SAI simulations are new here. As a whole, I think this section would be easier to interpret if you introduced SSP2-4.5 and ARISE-SAI-1.5, and the reference period and temperature targets, first, and then explained that you're presenting four new simulations which only target global mean temperature.

*We clarify that ARISE-SAI-1.5 was presented in a previous paper, and that the new sets of SAI simulations only target the global-mean temperature. We also clarify that the SSP2-4.5 simulations are one of UKESM1's core simulations carried out as part of CMIP6 (Sellar et al. 2019). We have also added a reference column to table 1.*

● Lines 69-70, "the global mean surface temperatures value in UKESM1 exceeds its preindustrial value by 1.5K" - this needs a citation, since the presented temperature data in Figure 1 only goes back to 2020 (I assume these numbers, and the temperature targets, were first presented in Henry, et al. 2023).

*We have added a reference to Henry et al. 2023.*

● Table 1: I would add a line to include SSP2-4.5, which is a unique set of simulations you are considering, and in line with my comment above, you could add a "first presented in" column with a citation/source for each simulation, even if it just says "Henry, et al. (2023)" for SSP2-4.5 and ARISE-SAI-1.5 and "here" for the rest. These changes would make it easier for a reader who is less familiar with the literature to follow along.

*Thank you for the suggestion of adding a reference column in the table, that will indeed make things clearer. SSP2-4.5 is described in Sellar et al. 2019.*

● Figure 1: the plot needs to be labeled better - clearly, the y-axis is relative to the preindustrial, but this is not stated. Additionally, the dashed black line is probably shown to represent the reference/target temperature, but it is not included in the legend.
● Figure 1: I would really like to see the individual ensemble members represented on this plot, not just the ensemble mean.
● Figure 1: I would also like to see actual numbers for global mean temperature somewhere, not just relative to the reference period - right now, the numbers are buried in the text; adding a right-side axis with the actual temperature values in °C or K, or adding the target value as text next to the dashed black line, would help give the reader another point of reference.
● Figure 1: the target and ARISE-SAI-1.5 both in black is a bit confusing - I would either pick a different color for one of them, or make the dashed line for the target thinner to more easily differentiate.

*We have clarified the figure by changing the y-axis label, changing the colour of the target line, adding the standard deviation of the ensemble, and editing the figure caption and legend. We keep the absolute temperature values in the text as we don't think adding them to the plot would provide any additional useful information.*

● Line 82 - saying Figure 2 "summarizes the climate response" is a bit strong... climate response is a lot more than zonal mean surface temperature and precipitation.

*We edit to "some key features of the climate response".*

● Line 85, "the most latitudinally homogeneous AOD is achieved by injecting at 30N+30S" - is it? The ARISE-SAI-1.5 AOD looks pretty similar to me. Additionally, you should clarify that this is out of the five strategies you considered here, and that this isn't a blanket truth for SAI.

*We remove that sentence.*

● Line 90, "the most efficient injection strategies" - 60N+60S isn't that different from the other three; certainly, the difference between 60N+60S and any of the other off-equatorial strategies is much larger than the difference between equatorial and off-equatorial

*60N+60S is ~20Tg/yr compared to ~15Tg/yr for ARISE-SAI-1.5, 15N+15S, and 30N+30S, which is ~30% higher. Hence our statement is valid. Note the change in injection amount at the equator as discussed in the introduction to the response to reviewers.*

● Lines 93-94 - I disagree with this characterization as "optimal," for the same reasons as above

*Agreed, we change that sentence.*

● Lines 99-100, "The larger injection rates for 60N+60S, on the other hand, arise due to faster removal of aerosols when injected near the descending branch of the Brewer Dobson Circulation (BDC)" - this needs a citation or other evidence to support it. This is one plausible contributing factor, but no analysis of aerosol lifetime is presented; polar SAI could simply cool the planet less efficiently because you're mainly inducing forcing changes over a relatively smaller fraction of the planet, and you're also injecting in winter when there's no sunlight to reflect.

*We agree with the reviewer's comments. We add an analysis of the aerosol lifetime ("The average lifetime of the injected stratospheric aerosols is 0.90 +/- 0.019, 0.87 +/- 0.024, 0.73 +/- 0.0094, 0.59 +/- 0.011, 0.80 +/- 0.020 years for the EQ, 15N+15S, 30N+30S, 60N+60S, and ARISE-SAI-1.5 simulations respectively. Here, the stratospheric aerosol lifetime (yr) is calculated as the ratio of the anomalous stratospheric SO2 burden (Tg) to the injection rate (Tg/yr), averaged over the last 10 years of the simulations.") and edit as follows: "The larger injection rates for 60N+60S, on the other hand, arise due to faster removal of aerosols when injected near the descending branch of the Brewer Dobson Circulation (BDC), as evidenced by the shortest lifetime of stratospheric aerosols for 60N+60S. In addition, the scarcity of sunlight at high latitudes during parts of the year further reduces the overall cooling efficiency of the 60N+60S injection strategy. We note that this shortcoming can be overcome by injecting aerosols only in spring in each hemisphere (Lee21)."*

● Figure 2, all panels: again, I would really like to see individual ensemble members, not just ensemble mean

*We have edited figure 2 to include standard deviations of the ensemble members.*

● Figure 2a: does "AOD" refer to 550nm AOD in the stratosphere only, or the whole atmosphere?

*Stratosphere only, we edit the figure caption accordingly.*

● Figure 2a: we need to see at least some information about the baseline SSP2-4.5 AOD here. I would plot raw AOD instead of $\Delta$AOD, and show the SSP2-4.5 in yellow as well; if SSP2-4.5 AOD is so small that it wouldn't even show up on this graph (e.g., order 0.01 or lower), you could just omit it entirely, and say in the text or figure caption that the baseline is several orders of magnitude smaller than the changes under SAI.

*We have added the following to the text: "Note that the baseline stratospheric AOD under SSP2-4.5 is three orders of magnitude smaller than the changes under SAI."*

● Figure 2b: I would really like to see timeseries instead of 10-year averages. Additionally, there is no description of what the errorbars represent; however, I would rather just see individual ensemble members here too.

*We have changed figure 2b to show the time-series of the injection rate with shading representing +/- 1 std of the ensemble members. And we add the average +/- 1 std over the last ten years in the legend (as now specified in the figure caption).*

● Figures 2c and 2d: using a black line for both target and ARISE-SAI-1.5 is a bit confusing - please either change the color of one, or make the line for the target thinner to differentiate

*We change the colour of the dashed target line to the same colour as for figure 1.*

● Figure 2d: while zonal mean surface temperature changes are probably okay, I think precipitation would be much better shown as a map. I know this would take up a lot of space, but you have maps for each strategy in figures 4-6; I don't know if you are constrained by length limits or number of figures, but it might be better to have one figure for AOD and injection rates (perhaps merged with Figure 1), and have a second figure containing temperature and precipitation. If this isn't feasible, I encourage you to add maps of precipitation changes to the supplementary.

*We add maps of temperature and precipitation change to the supplementary figures, as we are trying to keep the main body of the paper relatively concise. (New figures A6 and A7).*

● Lines 110-111, "no pattern of AOD from SAI is able to entirely offset the forcing from greenhouse gasses in the model" - firstly, this paragraph discusses temperature change, not forcing; secondly, this is only shown for the five patterns considered here

*Yes, this is correct, we are explaining the temperature change by discussing the forcing pattern as that is a major driver (along with feedbacks) of the temperature change.*

*We edit the sentence as follows "This shows that no pattern of AOD from SAI is able to entirely cancel out the spatial forcing from greenhouse gases in the model. This is especially the case in the Arctic where greenhouse gases exert a longwave forcing year-round whereas no SAI aerosol shortwave forcing will occur during the polar winter.". This is true for all SAI methods as they act on the shortwave.*

● Lines 133-134 - more detailed methodology is needed here; are you interpolating between grid boxes to compute this? Is this a linear interpolation, or a cubic one?

*We edit the sentence as follows : "Here, the ITCZ is computed as the linear interpolation of the latitude near the equator where the zonal-mean mass streamfunction at 500 hPa changes sign."*

● Line 141 - I am not convinced this relationship is linear. You have fit a linear trend to the data, but that does not mean the relationship is linear
● Figure 3: how are you computing the dashed line? More details are needed here
● Figure 3: rather than using the bars to denote ensemble spread, I suggest just using different color dots for each of the individual simulations, as the number of individual simulations is relatively small

*We have substantially changed the structure of this paragraph and hope that the text is sufficiently clear about the complexities of the ITCZ location and that the linear relationship calculation serves to estimate how much the warming itself (in 2050-69 SSP2-4.5) brings the ITCZ towards the equator.*

*We add the following : "For the SAI simulations, there is a correlation between the latitude of the ITCZ and the hemispheric difference in temperature (T1) as shown in figure 3 (dashed line), which is estimated by fitting a line which minimises the least squared error. The linear function is as follows : ITCZ latitude = 11.8*T1 - 2.9. Based on this linear assumption, the predicted ITCZ latitude…"*

*Using dots gave the following plot, which we don't think is helpful.*

[Figure]

● Lines 156-158 - there is already at least some literature on this; see Lee, et al. (2020)

*Indeed, we add ", as was done in Lee et al. 2020." at the end of that sentence.*

● Figure 4 (and Figure 6) - height in km, or pressure in hPa, would probably be easier to read than height in m

*We change the height to km for figures 4 and 6.*

● Line 171 - if age of air cannot be measured directly, can you elaborate on how it is presented here? Do you compute it from the model output, or is it computed and saved directly as model output during the simulation?

*The first sentence clearly refers to atmospheric measurements of age of air. We add ", as output by the model, " to the following sentence.*

● Conclusion, in general: the results here are primarily compared to similar experiments in CESM2, but I am also curious about the robustness of these results in UKESM - for example, Wells, et al. (2024) looked at equatorial injection vs. a four-latitude controller strategy similar to ARISE-SAI-1.5, but with a different background; if that study presented any of the same diagnostics as this study (ITCZ position, circulation changes, ozone depletion), it would be a useful point of comparison.

*Good point, we add a comparison wrt to circulation changes, ozone impacts, and AOD increase.*
*"This is consistent with a marked reduction in the Hadley Circulation intensity (figure A3) and with findings from Wells et al. 2024 (their figure 7)".*
*"These changes are consistent with Wells et al. 2024 (their figure 9)." in section 3.2 when discussing ozone changes.*
*"The equatorial peak in AOD is also consistent with simulations presented in Wells et al. (2024) which had a different background scenario and target state."*

*And in the conclusion:*

*"To achieve the same temperature target, the strategy requires 14% more injection relative to 30N+30S, and results in large reductions in tropical precipitation and total column ozone in the tropics, a marked reduction in the Hadley Circulation intensity, and a large tropical lower stratospheric warming. These are all consistent with findings from Wells et al. 2024, which used UKESM1 but with a different background scenario and target climate."*

● Conclusion, in general: I like the objectivity of this section a lot more than the abstract/introduction; rather than saying that 30N+30S is the "best" strategy, evidence is presented that certain possibly unwanted impacts are smaller or absent. The equatorial strategy is still described as having "the most negative side effects" in lines 252 and 270, which I would advise rewording. Lastly, in line 272, I might choose a word such as "supports" instead of "Confirms".

*We make the suggested edits:*

*"Finally, the equatorial strategy leads to trapping of the aerosols inside the tropical pipe, thus resulting in the largest impacts on atmospheric temperatures and circulation. To achieve the same temperature target, the strategy requires 14% more injection relative to 30N+30S, and results in large reductions in tropical precipitation and total column ozone in the tropics, a marked reduction in the Hadley Circulation intensity, and a large tropical lower stratospheric warming."*

*We change "confirms" to "supports the idea that [...]".*

**Reviewer 2**

This study compares the climate response to stratospheric aerosol injection in the earth system model UKESM under varying injection latitude. It is a clear presentation of an important new set of simulations, which will be a useful reference going forwards. It makes a valuable contribution in clarifying the trade-offs between different possible injection latitudes. I would like to see it published given some minor revisions.

General comments:

In the abstract, the statement "*we demonstrate that the 30N+30S strategy has, on balance, the least negative side effects*" is too strong. There is no comprehensive assessment of the side effects or their impacts presented here, and their results suggest 60° injection has smaller side-effects on some metrics. I suggest the authors make a more limited statement, or at least specify on what metric they are making this judgement.

*That's a fair point, we change the last sentence of the abstract as follows: "Finally, while all the SAI strategies come with trade-offs, our work shows that the 30N+30S strategy is a good candidate strategy for an inter-model comparison, and is easier to implement than a multi-latitude controller algorithm which has been adopted by GeoMIP under the G6-SAI-1.5K strategy (Visioni et al., 2024)."*

It would be good to include maps of the global surface climate response, at least for temperature and precipitation, under the different scenarios. This could be in Supplementary if article length is an issue. While not critical to the arguments being made here, these simulations will be a useful reference point for others and I suspect many will want to see the full spatial picture.

*This is also a fair point. We add maps of temperature and precipitation change to the supplementary figures (figs A6 and A7), as we are trying to keep the main body of the paper relatively concise.*

The zonal line plots (fig 1, fig 2a,c,d) don't currently indicate the ensemble spread. This information would be useful to include (perhaps via a new figure in Supplementary if it is too cluttered here), since it's not currently obvious where the differences between scenarios are robust to internal variability.

Fair point, the zonal line plots have all been edited to include the ensembles' standard deviation in shading.

Figure 2C: there are two aspects here which could be a little misleading, and which adding some more subplots (or a new fig in supplementary) could resolve. First, the equatorial injection strategy looks as though it strongly undercools the arctic relative to the other strategies, but this is partly an artifact of the controller failing to meet the 1.5K target for this strategy. Perhaps also plotting the residuals for each strategy relative to the baseline global mean temperature it

actually achieves (e.g. relative to a ~1.75K world for equatorial injection) would show more clearly the relative differences in latitudinal pattern of cooling between the strategies. Second, plots like these obscure the lower latitude structure because of the strong feedback driven arctic amplification of residuals (as already discussed in the text). Perhaps it would be worth also plotting the residual warming scaled by the local warming from pre-industrial at the baseline?

*Those are great points, we include the global temperature change of the last 20 years for each ensemble in figure 1, and include a supplementary figure with 2 panels showing the temperature factored by the global-mean temperature change, as well as the zonal-mean temperature change excluding the Arctic (North of 60N). And we add to the discussion of fig 2: "We note that the large Arctic temperature change hides the pattern of surface temperature change elsewhere in figure 2c, hence Figure A2(a) shows the temperature change excluding the Arctic region. Additionally, the smaller amount of cooling for the EQ strategy (fig 1) may exaggerate the undercooling of the Arctic. Therefore Figure A2(b) shows the surface temperature change for the SAI strategies relative to SSP2-4.5 normalized by the global-mean temperature change."*

Minor points:

Line 95: "*The larger injection rate is thus due to the lower efficacy of tropical forcing (Kang and Xie, 2014) and to the confinement of aerosols inside the tropical pipe, enhancing the formation of larger aerosols which sediment faster.*" A third potential contribution is that local forcing is sub-linear in local AOD, so concentrating AOD in one region reduces the global mean forcing.

*I am not sure this is accurate. If we compare figs 2 and 7 of Visioni et al. (2023) (Climate response to off-equatorial stratospheric sulfur injections in 3 ESMs…, https://doi.org/10.5194/acp-23-663-2023), UKESM1 also has an AOD peak when injecting at the EQ (fig 2c), but the cooling per AOD is not that different between injection locations (fig 7).*

*Also, note the factor of 2 change in the EQ injection amount relative to the first submission. This was due to a data processing error so only affects fig 2b.*

Figure 2b: define the error bars here. Are they the ensemble spread?

*We changed figure 2b to include a timeseries of the injection rate (with the standard deviation).*

Line 99: "*The larger injection rates for 60N+60S, on the other hand, arise due to faster removal of aerosols when injected near the descending branch of the Brewer Dobson Circulation (BDC)*". There are other potential contributions here; latitudinal even-ness of the AOD distribution (as above), and latitudinal variation in insolation, underlying albedo, and feedbacks

*As above, we don't think the latitudinal even-ness argument is valid here.*

*We edit this section as follows : "The larger injection rates for 60N+60S, on the other hand, arise due to faster removal of aerosols when injected near the descending branch of the Brewer Dobson Circulation (BDC), as evidenced by the shortest lifetime of stratospheric aerosols for 60N+60S. In addition, the scarcity of sunlight at high latitudes during parts of the year further reduces the overall cooling efficiency of the 60N+60S injection strategy. We note that this shortcoming can be overcome by injecting aerosols only in spring in each hemisphere (Lee21)."*

Label text is too small on the multi-panel figures (4,5,6)

*Fixed.*

Line 227-228 perhaps rephrase to make it clear that it is in CESM2 that the high latitude runs were seasonal and lower altitude.

*We add "in CESM2" as follows : "It is inspired from a similar study using CESM2 (Zhang et al., 2024) with the only differences being that the high-latitude injections **in CESM2** were done at a lower altitude and only in the spring of each hemisphere"*

Line 238: "*The main takeaway is that the 30N+30S strategy is one of the most efficient strategies in terms of amounts of SO2 needed".* Perhaps better to give a rough statement of the relative efficiencies. There aren't many strategies compared, so "one of the best" is not all that meaningful. My interpretation is 30° is only slightly less efficient (5%?) than the best efficiency strategy tested of 15°.

*We edit as follows : "The main takeaway is that the 30N+30S strategy is the second most efficient strategy among those presented in this manuscript in terms of amounts of SO2 needed (12% more injection than the most efficient 15N+15S strategy)."*

Line 249: perhaps worth stressing this point more strongly - given a strong uplift in efficiency for seasonal strategies (and a stronger uplift at 60° than at 30°) its not implausible that 60 seasonal would be the most efficient strategy in UKESM.

*Agreed. We edit as follows : ", and may plausibly make the 60N+60S strategy more efficient than the 30N+30S strategy in UKESM1" to that sentence.*

Figure 3: I wonder if it would be worth adding some more data to the plot to better explore the drivers. Adding at least the model's pre-industrial range might help to inform the discussion around lines 143-144 of the expected changes under warming.

*We include a supplementary figure (fig A8) showing the correlation between T1 and ITCZ in a historical run (reproduced below). Since the values are well out of the range of the current figure and have quite a large variation, we think it makes more sense to have it in a supplementary figure rather than in fig 3.*

*We have substantially edited this section for clarity.*

*And we add the following:*

*" We further note that a similar relationship inferred from the UKESM1 historical simulation suggests that a 0.4K change in T1 is needed to induce a 0.8 change in ITCZ latitude (fig A8). (This was not included in figure 3 for clarity purposes.) As such values also do not fit the relationship inferred from the SAI simulations above, the results highlight that factors other than the inter-hemispheric temperature gradient alone are important in modulating the ITCZ position. Thus, further developments of the controller might benefit from utilising more sophisticated metrics than a simple measure of interhemispheric temperature gradient to refine injection strategies, as has been demonstrated in Lee et al. (2020)."*

[Figure]

***Fig A8***

**Reviewer 3**

Summary :

This work used UKESM1 to perform a set of SAI SO2 injection strategies, designed to be easily compared with previous work performed with CESM2-WACCM. A single simulation was performed, following the experimental design of the ARISE-SAI experiment, previously applied within CESM2; which uses a feedback algorithm to meet multiple surface objectives by controlling yearly SO2 input at 15 N, 15S, 30N and 30S. The additional four simulations performed sought to isolate the impacts of the latitudinal location of injection on the climate response to SAI and separately simulated injections at the equator, 15N/15S, 30N/30S, and 60N/60S; fixing the altitude and season of injection. Taken together, these five simulations:

1. Enabled comparison between the climate response to SAI in UKESM1 and CESM2-WACCM to understand the ability to extrapolate conclusions drawn from single model studies
2. Enabled understanding of how injection latitude impacts the climate response to SAI in UKESM1

In general the response in UKESM was found to be similar to that of CESM2, apart from marked differences in the shifting of the ITCZ, motivating future study of the differences in the mechanistic controls on this feature between models. Comparing between the fixed latitude injections in UKESM, injection at 30N and 30S required the smallest amount of injected SO2 to reach set targets, as compared with other injection locations, and minimized undesirable side effects. Injection at the equator was found to be the least efficient with the largest undesired modifications to features like ozone and dynamics. Together these findings motivate the 30N/30S injection strategy as part of the GeoMIP G6-1.5K-SAI simulations, and suggest that the robustness climate impacts, like shifting of the ITCZ would benefit from an intermodel study.

General Comments:

In general, the narrative of this work could be strengthened by providing more introductory context. In particular, providing greater background pertaining to the feedback algorithm and how that simulation is important in the context of this work (e.g. the arise simulation does not isolate the latitude of injection, so why are you using it?). It is eventually clear that this work uses the ARISE simulation to ground comparison between CESM and UKESM, as well as to motivate the use of the 30N/30S injection for use in GeoMIP, but this narrative could be clearer from the introduction to set the reader up. Line by line suggestions to this point are listed below under "minor comments."

Additionally, language in the abstract is a slightly misleading; specifically "we demonstrate that the 30N+30S strategy has, on balance, the least negative side effects…" while in the context of the remainder of this paper the reader can understand this line, it overstates the depth of the analysis of the side effects. If the word count allows this should cite the specific effects analyzed and could be reworded to emphasize this conclusion motivates future simulation, versus future deployment, strategy.

*Thank you for your feedback and comments.*

*We agree with all reviewers that the language in this sentence is a little too strong as 30N+30S is not necessarily better than e.g. 60N+60S for all metrics. Hence, we change the last sentence of the abstract as follows: "Finally, while all the SAI strategies come with trade-offs, our work shows that the 30N+30S strategy is a good candidate strategy for an intermodal comparison, and is easier to implement than a multi-latitude controller algorithm."*

*We address your specific comments below.*

Minor Comments:

Line 20: "enables a control …" – should be more descriptive of the algorithm applied, and more clear that this is a PID controller seeking to optimize a set of multiple objectives

*We don't think this level of detail is necessary in the 3rd sentence of the introduction. Additionally, we cite 3 papers which contain much more detail about the controller.*

Line 31: "The contribution of the aerosol lifetime effects was found to be five to six times larger than that of the water vapor feedback." Unclear exactly what the aerosol lifetime is contributing to; and exactly how. In general it would be useful for the author to define "efficiency" as used in this paper – e.g. cooling per tg SO2 injected? Total reduction of the TOA imbalance per Tg SO2 injected? This should be directly defined in the introduction

*Agreed, we add "(as measured by the amount of cooling per Tg SO2 injected)".*

Line 40: How did Bednarz et al. vary cooling? Was it an idealized study or does this equate to a reduction of injected SO2?

*We clarify by adding "to maintain temperatures at 0.5 to 1.5 degrees above preindustrial temperatures".*

Line 42: The paragraph beginning with "looking into the future.." Felt out of place as is. This seemed disjointed from the logic flow and felt it could be better situated in the conclusions. It could be more useful to frame the introduction to motivate the use of the ARISE setup in UKESM as a way to compare the complex feedback simulation to the other four simulations – if one were to frame the paper this way, it would be important to emphasize the similarity of the conclusions between the ARISE and fixed latitude injection scenarios (e.g. does arise essentially recommend a 30N/30S injection? Are the climate outcomes about the same?).

*One of the key motivations of this work is to look at whether the 30N+30S strategy is a good choice for the next GeoMIP experiment (G6-SAI-1.5K). This is why we include the "Looking into the future…" paragraph. The structure of the introduction is 1/ intro to SAI and non-optimality of equatorial injection, 2/ dependence of climate outcomes on strategy (latitude, altitude, amount), 3/ G6-SAI-1.5K will be using 30N+30S which motivates usefulness of this paper, 4/ outline of the paper. We think this flows fairly well as-is.*

Line 72-75: "unlike ARISE-SAI …" : When defining the simulations performed it would be useful to clarify what season these injections are performed during; or rather are they continuous? Might considering adding injection timing as a column in Table 1.

*We clarify this by editing L.75 as follows : "All SAI simulations use SSP2-4.5 as their background greenhouse gas emission scenario and inject aerosols continuously throughout the year."*

Line 83: "…anomalous AOD..." the meaning of this is relatively unclear. Is this the AOD anomaly taken as the deseasonalized and detrended AOD ?

*We change "anomalous" to "the change in" and clarify that we are looking at the 2050-69 average.*

Line 90: Would be useful to formally define efficiency.

*We add "in terms of cooling per Tg SO2".*

Line 99: Would be interesting to include the values for the differences in volume-mean or mass-mean particle size between injection locations!

*We plot the mean particle volume as defined by the ratio of the mass mixing ratio and the number concentration times the density of sulfate below. But we do not add it to the manuscript as we don't think it adds much more relative to the effective radius (fig A1).*

[Figure]

Line 136: There is an expected relationship between ITCZ and the hemispheric temperature difference, this relationship is shown by the dashed line in figure 3, with a calculated slope of X. It was challenging to follow this paragraph, because to the reader it says the ITCZ is controlled by the hemispheric temperature difference, but actually it isn't … I would try to be more direct in this paragraph and lead with the many controls on the ITCZ, which explain why the linear correlation is a relatively bad fit. It would also be useful to compare the SAI-ARISE CESM2 results to the UKESM results here – especially given this a main point in the conclusion. Could include in Figure 3 to give context to the reader.

*We agree and follow your suggestion to restructure this paragraph completely to clarify the controls on the ITCZ, and explain why we plot the ITCZ as a function of T1. We hope that makes things clearer. (We do not copy this section below because of the length.)*

*We do not include CESM2 values here as it would make things too complicated but we discuss the comparison with CESM2 in the conclusion.*

Figure 4/5/6: When using a two sided ttest on spatially resolved data, the p value must be corrected for the false discovery rate. Refer to: http://dx.doi.org/10.1175/BAMS-D-15-00267.1

*Thank you for pointing that out! We now adjust our p-values for the false discovery rate thus increasing the p values and slightly increasing the amount of stippling, and mention it in the caption of the relevant figures.*

Line 169: "… causes a weakening of the tropospheric jets, …" I was curious what the seasonal (DJF, MAM, SON etc) figures look like for the zonal wind changes. It's not immediately clear what season the injections were performed during, however, if one expects to strengthen the stratospheric polar vortex, based on previous studies, this should delay the springtime breakdown of the vortex, resulting in a poleward shift of the eddy driven tropospheric jet. I wonder if the weakening seen here, might appear to be a poleward shift if parsed by season vs an annual average change for the given period.

*We have clarified that the injection is continuous year-round. We parse the plot by season below, and while a consistent weakening of the subtropical tropospheric jet can be seen for all seasons, the reviewer is correct in that in certain seasons the stratospheric westerly response can propagate down to the troposphere in the form of a poleward shift of eddy-driven jet; this can see for instance in the SH in austral summer (DJF) under EQ injection.*

*We have rephrased the end of the first paragraph in Section 3.2 to read: "In the troposphere, all SAI strategies simulate the largest cooling in the tropical upper troposphere (Figure A9); this causes a year-round weakening of the subtropical jets, again with the largest changes for the equatorial injection. In the extratropics, stratospheric westerly responses can at certain times propagate down to the troposphere below in the form of a poleward shift of the eddy-driven jet (e.g. Bednarz et al., 2023); a suggestion of such a response is for instance found in the southern hemisphere summer under the equatorial injection (not shown)."*

[Figure]

Conclusions:

+  Might consider a discussion of how season of injection (e.g. performing the same simulations in a different season) might change the outcomes.

*That has been explored in CESM2 but is not the focus of this study as it adds another layer of complexity, as would, for example, exploring how the height of injection might impact climate outcomes. We add the following to the introduction : "Finally, previous work has shown that changing the season of injection may impact regional climate outcomes (Visioni et al. 2019), and the efficiency in cooling per TgSO2 is increased when injection is limited to spring when injecting at high Northern latitudes and at 15km (Lee et al. 2021)."*

+ could be strengthened by a discussion of the known differences in UKESM and CESM2 alongside a more detailed discussion of the differences in response between the models. This was a major motivation of the investigation and does not feel fully fleshed out.

*We already compare our results with CESM2 both throughout the manuscript and in a whole paragraph in the conclusion, but we add more detail as follows:*

- *"The main takeaways are that the 30N+30S strategy is the second most efficient strategy among those presented in this manuscript in terms of amounts of SO2 needed (12\% more injection than the most efficient 15N+15S strategy), and is among the strategies which have the smallest changes in precipitation, position of intertropical convergence zone, ozone concentrations and atmospheric circulation (both in the troposphere and stratosphere), which is broadly consistent with previous results using CESM2."*
- *"This is different to CESM2 which has more than 1K extra Arctic cooling for their polar strategy, though the injection takes place at a lower altitude relative to UKESM1 and in the spring of each hemisphere. It is also worth noting that Arctic amplification is much less pronounced in CESM2 relative to UKESM1."*
- *The whole 3rd paragraph of the conclusion is about comparisons with CESM2.*